# Revenue Guarantees of No-Swap-Regret Dynamics in First Price Auctions

**Anders Bo Ipsen** [1]  **Stratis Skoulakis** [1]

## Abstract

We study the revenue of approximate correlated equilibrium in discrete first price auctions - the set of allowable bids is $\mathcal{B} = \{0, 1/k, \ldots, 1-1/k, 1\}$ for some $k \in \mathbb{N}$. We show that the revenue of any $\epsilon$-*approximate* correlated equilibrium is at least $v_2 - \Theta(1/k) - \Theta(\epsilon k^2)$, where $v_2 \geq 0$ is the second-highest valuation. Our results establish the first polynomial convergence rates on the revenue generated by no-swap regret bidders in first-price auctions. For instance, if bidders admit the optimal swap regret of $\mathcal{O}(\sqrt{kT})$, then the time-averaged revenue is at least $v_2 - \Theta(1/k) - \Theta(\epsilon)$ after $\mathcal{O}(k^5/\epsilon^2)$ rounds.

## 1. Introduction

Ad auctions, where firms compete for advertisement spots, constitute a multi-billion dollar industry (129 billion in the U.S. alone in 2019 (Goke et al., 2022)). Based on fundamental results in auction theory (Vickrey, 1961), the second-price auction format had been widely adopted. However, in recent years, there has been a significant shift from second-price to first-price auctions (Goke et al., 2022; Despotakis et al., 2021; Paes Leme et al., 2020). For instance, Google Ad Manager transitioned to first-price auctions in March 2019 (Goke et al., 2022).

In the past, second-price auctions were favored over first-price auctions for a simple reason: in a second-price auction, bidding *truthfully* is a dominant strategy for every bidder (Vickrey, 1961). This property made second-price auctions easier both to predict and analyze. In contrast, first-price auctions require bidders to deviate from straightforward bidding strategies, introducing greater complexity and unpredictability both for the bidder and the auctioneer.

The rise of machine learning and automated bidding strategies has significantly influenced bidding behavior. Even in first-price auctions, a bidder can leverage machine learning algorithms in order to adjust its bids to the bidding behavior of its competitors (Weed et al., 2016; Nekipelov et al., 2015). In particular, the online learning framework (Hazan, 2019) provides such decision-making algorithms, which also come with strong optimality guarantees for any bidder who adopts them. In particular, if an auction is repeated over $T$ rounds[1], a bidder can use such online learning algorithms to select its bids and be guaranteed that *no matter the behavior of the competing bidders, its overall utility approaches the utility of the best fixed bid!* The latter guarantees are known as *no-regret guarantees* and in recent years there has been a growing interest for developing such *no-regret algorithms* in various auction settings (Weed et al., 2016; Han et al., 2020; 2025; Wang et al., 2023; Feng et al., 2018; Nedelec et al., 2019; Karaca et al., 2020; Achddou et al., 2021; Potfer et al., 2024; Brânzei et al., 2023; Balseiro et al., 2023; Kumar et al., 2024; Bichler et al., 2023). Moreover, (Nekipelov et al., 2015; Noti & Syrgkanis, 2021) showed that online learning algorithms provide an accurate model for real-world bidding behavior.

For the reasons outlined above, a recent line of research investigates the revenue generated in first-price auctions when all bidders repeatedly choose their bids according to online learning algorithms (Dütting et al., 2014; Feldman et al., 2016; Feng et al., 2021; Kolumbus & Nisan, 2022; Deng et al., 2022; Bichler et al., 2023; Brânzei et al., 2023). In particular this line of works tries to characterize the limiting behavior of such the *bid dynamics* as the number of rounds approaches infinity. However, in practice, the number of rounds is indeed large (ad auctions occur every millisecond) yet finite. This naturally leads to the following question:

**Question 1.1.** *If in a repeated first-price auction, all bidders use online learning algorithms, how many rounds are needed so that the revenue reaches an acceptable level?*

*Remark* 1.2. Question 1.1 is of great importance when designing auction at which learning algorithms compete. Even if the produced revenue converges to a very high value but a tremendous number of rounds is required before this happens, the auctioneer might never receive this revenue exactly because the time horizon was not sufficiently long.

---

[*]Equal contribution [1]Department of Computer Science, Aarhus University, Denmark. Correspondence to: Anders Bo Ipsen <abi@cs.au.dk>, Stratis Skoulakis <stratis@cs.au.dk>.

*Proceedings of the 43rd International Conference on Machine Learning*, Seoul, South Korea. PMLR 306, 2026. Copyright 2026 by the author(s).

[1]Ad auction are repeated over a large number of rounds since firms repeatedly compete for the same keyword; such an auction can be repeated even every millisecond.

Our work is motivated by Question 1.1 that to the best of our knowledge has not been previously considered. In particular, we investigate Question 1.1 in the case of *discrete first-price auctions* and for online learning algorithms that admit *no-swap regret guarantees* (Blum & Mansour, 2007). Discrete first-auction capture settings where bidders can only submit specific discretized bids - the set of allowable bids is the discretized $[0, 1]$ interval, $\mathcal{B} := \{0, 1/k, \ldots, 1 - 1/k, 1\}$). No-swap regret algorithms are online learning algorithms that provide even stronger optimality guarantees than standard *no-(external) regret* algorithms. No-swap regret algorithms have been extensively studied to date (Anagnostides et al., 2022a; Daskalakis et al., 2024; Farina & Pipis, 2024; Zhang et al., 2024b; Dagan et al., 2024; Zhang et al., 2024a; Peng & Rubinstein, 2024) while their time-average behavior is known to be a *correlated equilibrium* (Aumann, 1974).

## 1.1. Correlated Equilibrium & No-Swap Dynamics

Through some very elegant arguments, (Dütting et al., 2014; Feldman et al., 2016) were able to characterize the set of *correlated equilibria* in first-price auctions (limiting behavior of no-swap regret algorithms). In particular their results imply the following:

**Informal Theorem** *Let a discrete first-price auction where $v_2 \in \mathcal{B}$ is the 2nd highest valuation. Then the revenue of any correlated equilibrium[2] is at least $v_2 - \Theta(1/k)$.*

Since the time-average behavior of no-swap regret algorithms always converges to a correlated equilibrium (Foster & Vohra, 1997; Hart & Mas-Colell, 2000), the result above suggests the following:

**Claim 1.3.** *Let a discrete first-price auction that is repeated for $T$ rounds and bidders use no-swap regret algorithms. Then as $T \to \infty$, the time-average revenue is at least $v_2 - \Theta(1/k)$.*

Claim 1.3 suggests that as $T \to \infty$, the revenue produced by no-swap regret dynamics approaches the second highest valuation since $v_2 - \Theta(1/k) \simeq v_2$ for a sufficiently large discretization $k \in \mathbb{N}$. However, it does not provide any upper bound on the required number of rounds $T$ for the latter to happen. This is because, for finite $T$, no-swap regret dynamics converge to *approximate* correlated equilibrium, whereas the results of (Dütting et al., 2014; Feldman et al., 2016) concern *exact* correlated equilibria.

With some effort, the arguments of both (Dütting et al., 2014) and (Feldman et al., 2016) can be extended to approximate correlated equilibria. However, the revenue guarantees then exhibit a quasi-polynomial dependence on $k$. In

particular, by extending the arguments from (Feldman et al., 2016), one can establish the following[3].

**Theorem 1.4.** *The expected revenue of an $\epsilon$-approximate correlated equilibrium in a discrete first-price auction is at least $v_2 - \Theta(1/k) - \epsilon \cdot k^{\text{poly}(\log k)}$.*

On the positive side, the latter result can be used to provide an upper bound on the number of rounds $T$ needed so that the revenue, produced by competing no-swap regret algorithms, exceeds $v_2 - \Theta(1/k)$. On the negative side, the provided upper bound on $T$ is extremely large with respect to $k$. In particular, even if all bidders use the optimal algorithm from (Blum & Mansour, 2007) with swap regret $\mathcal{R}(T) := \mathcal{O}(\sqrt{kT})$, the upper bound on $T$ becomes $k^{\mathcal{O}(\log k)}$. The latter raises the following natural question.

**Question 1.5.** *Does any $\epsilon$-approximate correlated equilibrium of a discrete first-price auction, admits revenue that is at least $v_2 - \Theta(1/k) - \epsilon \cdot \text{poly}(k)$?*

## 1.2. Our Contribution and Results

In this work we answer Question 1.5 on the affirmative. In particular, our main result is the following:

**Theorem 1.6.** *The expected revenue of an $\epsilon$-approximate correlated equilibrium in a discrete first-price auction is at least $v_2 - \Theta(1/k) - \Theta(\epsilon \cdot k^2)$.*

As already discussed, our result then translates to a way faster rate on the time-average revenue produced by no-swap regret bidders in first-price auctions.

**Claim 1.7.** *If each bidder uses an online learning algorithm with swap regret at most $\mathcal{R}(T)$. Then after $T$ rounds, the time-average revenue is at least $v_2 - \Theta\left(\frac{1}{k}\right) - k^2 \frac{\mathcal{R}(T)}{T}$. If $\mathcal{R}(T) = \mathcal{O}(\sqrt{kT})$ then at most $T := \mathcal{O}\left(k^5/\epsilon^2\right)$ rounds are needed before the time-average revenue is at least $v_2 - \Theta(1/k) - \Theta(\epsilon)$.*

**Additional Results:** Our main result that provides a positive answer to Question 1.5 is formally stated and proven in Theorem 2.6. In Section 3, we present the key steps for establishing it. In Appendix A, we demonstrate how the elegant argument of (Feldman et al., 2016) that concerns *exact* correlated equilibrium in *continuous first-price auctions* can be easily extended to discrete first-price auctions. We also sketch how it can be extended to approximate correlated equilibrium with a quasi-polynomial bound. The latter is done only since Theorem 2.6 provides a polynomial bound. Apart from lower bounds on the expected revenue of approximate correlated equilibrium, we also present an upper bound on the revenue. In particular in Theorem 2.8 we establish that there is a simple first-price that

---

[2]The original results of (Dütting et al., 2014; Feldman et al., 2016) concern first-price auctions with continuous bidding space $[0, 1]$ or equivalently $k \to \infty$. However their arguments can be easily transferred to the discrete setting.

[3]The arguments in (Dütting et al., 2014) can also be extended to approximate correlated equilibria, but they provide a weaker guarantee of $v_2 - \mathcal{O}(1/k) - \epsilon \cdot k^{\text{poly}(k)}$

admits an $\epsilon$-approximate correlated equilibrium with revenue $v_2 - 1/k - \epsilon$. In Section 4, we experimentally evaluate the no-swap regret algorithm of (Blum & Mansour, 2007) in the context of first-price auctions. Our experimental evaluations concern both the case of deterministic evaluations as well as i.i.d. and suggest that the resulting dynamics converge even faster than what our analysis suggests.

**Our Techniques:** In order to answer Question 1.5, we adopt a completely different approach from both (Dütting et al., 2014) and (Feldman et al., 2016). In particular, we adopt a *dual-fitting approach* - a technique used in approximation algorithms (Williamson & Shmoys, 2011). To the best of knowledge, our work is the first to introduce the dual-fitting technique in the context of establishing convergence rates and may be of independent interest. Our approach begins by introducing an appropriate linear program that, given an instance of a discrete first-price auction (with bidders' valuations and discretization parameter $k$), its optimal value (denoted by $Z_{\text{LP}}^\star$) is a lower bound to the revenue of any $\epsilon$-approximate correlated equilibrium. Next, by exploiting the structure of the dual program, we provide a specific assignment to the dual variables. We then establish that this assignment is dual feasible and admits objective value $v_2 - \Theta(1/k) - \Theta(\epsilon \cdot k^2)$. Thus, by weak duality we get that $Z_{\text{LP}}^\star \geq v_2 - \Theta(1/k) - \Theta(\epsilon \cdot k^2)$

### 1.3. Related Work

Apart from the works of (Dütting et al., 2014; Feldman et al., 2016), our work is closely related to a recent research line on the convergence properties of no-regret algorithms in discrete first-price auctions (Feng et al., 2021; Kolumbus & Nisan, 2022; Deng et al., 2022; Ahunbay & Bichler, 2025). Specifically, (Feng et al., 2021) provide positive results on the convergence of no-regret dynamics in discrete first-price auctions, assuming that all agents undergo an initial exploration phase. (Kolumbus & Nisan, 2022) demonstrate that if no-regret dynamics in first-price auctions converge, they essentially converge to the second-highest price, and they provide experimental evidence supporting this convergence. (Deng et al., 2022) establish that *mean-based no-regret algorithms* asymptotically converge in discrete first-price auctions under the assumption that there are at most two bidders with the highest valuation. (Ahunbay & Bichler, 2025) shows, using a strong-duality argument, that *projected gradient ascent* dynamics eventually converges to a Nash equilibrium under some weak assumptions on the game dynamics. Our work distinguishes itself by considering no-swap regret bidders and provides unconditional convergence results for all no-swap regret dynamics while providing explicit convergence rates.

Closely related to our work is also (Banchio & Skrzypacz, 2022), which experimentally analyze the revenue produced

by Q-learning algorithms in first-price auctions. At the same time, (Syrgkanis & Tardos, 2013), (Lykouris et al., 2016), and (Hartline et al., 2015) provide guarantees on the *social welfare* generated by no-regret dynamics in Bayesian games, including first-price auctions. (Balseiro & Gur, 2017) examine the convergence properties of no-regret algorithms in repeated auctions with budgets. (Daskalakis et al., 2020) establish that designing no-regret algorithms for repeated combinatorial auctions is a computationally intractable task. Finally a recent line of works studies equilibria and dynamics in autobidding systems (Leme et al., 2024; Fikioris & Tardos, 2023; Liu & Shen, 2023; Li & Tang, 2024).

## 2. Preliminaries and Notation

### 2.1. First-Price Auctions

We study (discrete) first-price auctions with $n$ bidders and a discrete set of possible bids $\mathcal{B} = \{0, \frac{1}{k}, \ldots, 1 - \frac{1}{k}, 1\}$. Each bidder $i \in [n]$ has a private valuation $v_i$ and submits bid $s_i$ which is bounded by their private valuation, specifically $s_i \in \mathcal{S}_i$ where $\mathcal{S}_i = \{0, \frac{1}{k}, \ldots, v_i\}$. We let $\mathcal{S} := \times_{i \in [n]} \mathcal{S}_i$ and then denote $s = (s_1, \ldots, s_n) \in \mathcal{S}$ (or $s = (s_i, s_{-i})$) a bidding profile. For a given bidding profile, the highest bidder $i^\star \in \arg \max_{j \in [n]} s_j$ wins but in the case of ties the winner is chosen *uniformly at random* amongst the highest bidders. This choice of tie-breaking rule motivates our notion of (expected) utility, captured in Definition 2.1.

**Definition 2.1.** Let a bidding profile $s = (s_1, \ldots, s_n) \in \mathcal{S}$. Then, the utility of agent $i \in [n]$ is

$$U_i(s_i, s_{-i}) := (v_i - s_i) \cdot \frac{\mathrm{I}\left[s_i = \max_{j \in [n]} s_j\right]}{|\{k : s_k = \max_{j \in [n]} s_j\}|}$$

### 2.2. No-Swap Regret Minimization

Let a first-price auction be repeated for $T$ rounds as described in Algorithm 1. This means that at each round, all bidders simultaneously select their bids and the bidder with highest bid wins the item.

---

**Algorithm 1** Repeated First-Price Auction

At each round $t = 1, \ldots, T$

- Each bidder $i \in [n]$, (secretly) selects a bid $s_i^t \in \mathcal{S}_i$.

- Each bidder $i \in [n]$, gets utility $U_i(s_i^t, s_{-i}^t)$.

---

In most natural cases a bidder $i \in [n]$ has no information on the valuations $v_{-i}$ of the other bidders. Thus, in order to maximize its utility, the bidder $i \in [n]$ must have a *bidding strategy* that in each round $t \in [T]$ depends solely on previous bids $s_1, \ldots, s_{t-1} \in \mathcal{S}$. The online learning framework provides such adaptive decision-making algorithms

that base their decision on prior observations (Hazan, 2019). In the context of first-price auctions, an online learning algorithm $\mathcal{A}$, in each round $t \in [T]$ produces a mixed strategy $x_i^t \in \Delta(\mathcal{S}_i)$ that depends solely on the previous bids of the other bidders $s_{-i}^1, \ldots, s_{-i}^{t-1} \in \mathcal{S}_{-i}$.

The performance of an online learning algorithm $\mathcal{A}$ can be quantified through various notions (Hazan, 2019). One of the most fundamental ones has been that of *swap regret* (Blum & Mansour, 2007). In particular the swap regret of $\mathcal{A}$, denoted as $\mathcal{R}_{\mathcal{A}}^{\text{swap}}(T)$, equals

$$
\begin{aligned}
\mathcal{R}_{\mathcal{A}}^{\text{swap}}(T) \quad := \quad & \max_{\delta:\mathcal{S}_i \mapsto \mathcal{S}_i} \sum_{t=1}^{T} \mathbb{E}_{s_i \sim x_i^t} \left[ U_i(\delta(s_i), s_{-i}^t) \right] \\
& - \sum_{t=1}^{T} \mathbb{E}_{s_i \sim x_i^t} \left[ U_i(s_i, s_{-i}^t) \right]
\end{aligned} \tag{1}
$$

Swap regret quantifies the difference of the overall utility produced by a no-regret algorithm with respect to the utility produced by the *best switching function* $\delta(\cdot)$ that replaces action $s_i$ with the action $\delta(s_i)$. If an online learning algorithm $\mathcal{A}$ admits sublinear swap regret $\mathcal{R}_{\mathcal{A}}^{\text{swap}}(T) = o(T)$ then $\mathcal{A}$ is called *no-swap regret*. (Blum & Mansour, 2007) provide a no-swap regret algorithm $\mathcal{A}$ with $\mathcal{R}_{\mathcal{A}}^{\text{swap}}(T) = \mathcal{O}(\sqrt{|\mathcal{S}_i|T})$ that is also the optimal possible no-swap regret.

### 2.3. Correlated Equilibrium and No-Swap Dynamics

Correlated Equilibrium (CE) is a fundamental notion introduced by Aumman (Aumann, 1974).

**Definition 2.2.** *A probability distribution $\mu \in \Delta(S)$ over the set of bidding profiles $\mathcal{S} := \times_{i \in [n]} \mathcal{S}_i$, is an $\epsilon$-approximate correlated equilibrium if for each bidder $i \in [n]$,*

$$
\mathbb{E}_{s \sim \mu} \left[ U_i(s_i, s_{-i}) \right] \geq \max_{\delta:\mathcal{S}_i \mapsto \mathcal{S}_i} \mathbb{E}_{s \sim \mu} \left[ U_i(\delta(s_i), s_{-i}) \right] - \epsilon.
$$

It is well-known that correlated equilibria describe the limiting behavior of no-swap regret algorithms (Hart & Mas-Colell, 2000; Foster & Vohra, 1997). In Theorem 2.3, we state the folkore results that the limiting behavior of no-swap regret algorithms forms an approximate correlated equilibrium. For completeness its proof is presented in Appendix C.

**Theorem 2.3.** *If each bidder $i \in [n]$, selects its bid $s_i^t \in \mathcal{S}_i$ according to a no-swap regret algorithm $\mathcal{A}_i$, $s_i^t \sim x_i^t$. Then after $T = \mathcal{O}\left( \max_{i \in [n]} \mathcal{R}_i(T)/\epsilon + \log(n/\delta)/\epsilon \right)$ rounds, the time-average distribution $\hat{\mu} \in \Delta(\mathcal{S})$, defined as*

$$
\hat{\mu}(s) := \frac{1}{T} \sum_{t=1}^{T} \mathrm{I}[s_t = s] \quad \text{for all } s \in \mathcal{S},
$$

*is an $\epsilon$-approx. correlated equilibrium with probability $1 - \delta$.*

Theorem 2.3 informs us that if bidders bid according to a no-swap regret algorithm then the time-average behavior converges to an approximate correlated equilibrium with rate $\tilde{\mathcal{O}}\left( \max_i \mathcal{R}_i(T)/T \right)$.

**Definition 2.4.** *For a bidding profile $s \in \mathcal{S}$, we denote $r(s) := \max(s_1, \ldots, s_n)$.*

Using Definition 2.4, the expected revenue of a joint probability distribution $\mu \in \Delta(\mathcal{S})$, that equals $\mathbb{E}_{s \sim \mu}[\max(s_1, \ldots, s_n)]$, can be more concisely written as $\sum_{s \in \mathcal{S}} \mu(s) \cdot r(s)$.

### 2.4. Our Results and Paper Organization

As already mentioned, (Dütting et al., 2014; Feldman et al., 2016) establish that in case of continuous first-price auctions, $\mathcal{B} = [0, 1]$, the revenue of any correlated equilibrium equals $v_2$. In Appendix A, we show how the very elegant argument of (Feldman et al., 2016) can be almost directly transferred to discrete first-price auctions, leading to the following theorem.

**Theorem 2.5.** *Let the probability distribution $\mu \in \Delta(\mathcal{S})$ be an exact correlated equilibrium ($\epsilon = 0$). Then its expected revenue admits $\sum_{s \in S} \mu(s) \cdot r(s) \geq v_2 - 4/k$.*

In Appendix A, we additionally demonstrate how the approach of (Feldman et al., 2016) can be extended for $\epsilon$-approximate correlated equilibrium. In particular, the expected revenue of any $\epsilon$-approximate correlated equilibrium $\mu \in \Delta(S)$ is at least $\sum_{s \in \mathcal{S}} \mu(s) \cdot r(s) \geq v_2 - 4/k - \epsilon \cdot k^{\Theta(\log k)}$. We remark that this is done only for completeness since the latter result is also implied by our main result (Theorem 2.6).

In Section 3, we present the main proof of Theorem 2.6 that is the main result of this work.

**Theorem 2.6.** *Let the probability distribution $\mu \in \Delta(\mathcal{S})$ be an $\epsilon$-approximate correlated equilibrium. Then its expected revenue admits $\sum_{s \in \mathcal{S}} \mu(s) \cdot r(s) \geq v_2 - 3/k - \Theta(\epsilon \cdot k^2)$.*

Combining Theorem 2.6 with Theorem 2.3 we get the following corollary on the time-average payoff of a first-price auction once all bidders use no-swap regret algorithms.

**Corollary 2.7.** *If each bidder $i \in [n]$, selects its bid $s_i^t \in \mathcal{S}_i$ according to a no-swap regret algorithm $\mathcal{A}_i$, $s_i^t \sim x_i^t$. Then after $T = \mathcal{O}\left( k^2 \max_{i \in [n]} \mathcal{R}_i(T)/\epsilon + k^2 \log(n/\delta)/\epsilon \right)$ rounds, the time-average revenue admits*

$$
\frac{1}{T} \sum_{t=1}^{T} r(s_t) \geq v_2 - \frac{3}{k} - \mathcal{O}(\epsilon)
$$

*with probability $1 - \delta$. If all agents use the algorithm of (Blum & Mansour, 2007) with swap regret $\mathcal{R}_i^{\text{swap}}(T) := \mathcal{O}(\sqrt{kT})$ then at most $T = \mathcal{O}\left( k^5 \cdot \frac{\log(n/\delta)}{\epsilon^2} \right)$ rounds are required.*

We remark that the upper bound of $T = \tilde{\mathcal{O}}\left(k^5/\epsilon^2\right)$ represents a significant improvement over the previous bound of $T = k^{\mathcal{O}(\log k)}/\epsilon^2$. Furthermore, if all bidders adopt specific no-swap regret algorithms, such as those proposed in (Anagnostides et al., 2022b;a), which guarantee no-swap regret of $\tilde{\mathcal{O}}(n)$ and $\tilde{\mathcal{O}}(nk^4)$, respectively, then the required number of rounds reduces to $\tilde{\mathcal{O}}(k^2n/\epsilon)$ and $\tilde{\mathcal{O}}(k^6n/\epsilon)$, respectively.

We finally provide an upper bound on the revenue of any $\epsilon$-approximate correlated equilibrium, establishing that the $\mathcal{O}(\epsilon)$-dependence on cannot be alleviated from the revenue guarantees of an $\epsilon$-approximate correlated equilibrium. The latter is formally stated and proven in Theorem 2.8.

**Theorem 2.8.** *Let the discrete-price auctions with $2$ bidders with $v_1 = v_2 = 1$. There exists an $\epsilon$-approximate correlated equilibrium $\mu \in \Delta(\mathcal{S})$ such that the expected payoff admits*

$$\sum_{s \in \mathcal{S}} \mu(s) \cdot r(s) \leq 1 - \Theta(1/k) - \Theta(\epsilon)$$

The proof of Theorem 2.8 is presented in Appendix B.

# 3. Proof of Theorem 2.6

In this section we present the proof of Theorem 2.6 that is the main contribution of this work.

**Theorem 2.6.** *Let the probability distribution $\mu \in \Delta(\mathcal{S})$ be an $\epsilon$-approximate correlated equilibrium. Then its expected revenue admits $\sum_{s \in \mathcal{S}} \mu(s) \cdot r(s) \geq v_2 - 3/k - \Theta(\epsilon \cdot k^2)$.*

In order to establish Theorem 2.6 we develop a novel dual-fitting approach. In Definition 3.1 we introduce a linear program the minimum value of which acts as a lower bound on the revenue of any $\epsilon$-approximate correlated equilibrium.

**Definition 3.1.** *Let a discrete first-price auction and consider the following linear program,*

$$\min \sum_{s \in \mathcal{S}} \mu(s) \cdot r(s)$$
$$\text{s.t.} \sum_{s_{-i} \in \mathcal{S}_{-i}} \mu(s_i, s_{-i})\left(U_i(s_i', s_{-i}) - U_i(s_i, s_{-i})\right) \leq \epsilon$$
$$\quad \text{for all agents } i \in [n] \text{ and bids } s_i, s_i' \in \mathcal{S}_i$$
$$\sum_{s \in \mathcal{S}} \mu(s) = 1$$
$$\mu(s) \geq 0 \quad \forall s \in \mathcal{S}$$

*where $U_i(s_i, s_{-i})$ is the utility of agent $i \in [n]$. We denote with $Z_{LP}^\star$ the optimal value of the linear program.*

Notice that any feasible solution $\mu(\cdot)$ of the polytope described in Definition 3.1 is also a joint probability distribution over $\mathcal{S}$, $\mu \in \Delta(S)$. In Lemma 3.9 we establish that the optimal value $Z_{LP}^\star$ of the linear program in Definition 3.1 acts as a lower bound to the revenue of any $\epsilon$-approximate correlated equilibrium $\mu \in \Delta(\mathcal{S})$.

**Lemma 3.2.** *Let $\mu \in \Delta(\mathcal{S})$ be an $\epsilon$-approximate equilibrium. Then $\sum_{s \in \mathcal{S}} \mu(s) \cdot r(s) \geq Z_{LP}^\star$.*

The proof of Lemma 3.2 consists of using Definition 2.2 to show that any $\epsilon$-approximate correlated equilibrium satisfies the constraints of the linear program of Definition 3.1 and is relative straightforward (see Appendix D for the proof). We dedicate the rest of the section to establish a lower bound on $Z_{LP}^\star$. In Lemma 3.3 we present the dual of the linear program of Definition 3.1 (see Appendix D for the proof).

**Lemma 3.3.** *The dual of the LP in Definition 3.1 is the following:*

$$max \quad \mu - \epsilon \cdot \sum_{i \in [n]} \sum_{s_i \in \mathcal{S}_i} \sum_{s_i' \in \mathcal{S}_i} \lambda_{s_i s_i'}^i$$
$$s.t. \quad \mu + \sum_{i \in [n]} \sum_{s_i' \in \mathcal{S}_i} \lambda_{s_i s_i'}^i\left(U_i(s_i, s_{-i}) - U_i(s_i', s_{-i})\right) \leq r(s)$$
$$\quad \text{for all } s \in \mathcal{S}$$
$$\lambda_{s_i s_{-i}}^i \geq 0 \quad \forall i \in [n], s_i, s_{-i} \in \mathcal{S}_i$$

*Remark* 3.4. We remark that the dual variable $\mu \in \mathbb{R}$ in Lemma 3.3 is a scalar and should not be confused with $\mu(\cdot) \in \Delta(\mathcal{S})$ in the primal linear program of Definition 3.1.

Our dual fitting approach consists of finding an assignment to the dual variables $\{\mu, \lambda_{s_i s_i'}^i\}$ that is feasible for the dual and satisfies

$$\mu - \epsilon \cdot \sum_{i \in [n]} \sum_{s_i \in \mathcal{S}_i} \sum_{s_i' \in \mathcal{S}_i} \lambda_{s_i s_i'}^i \geq v_2 - 3/k - \Theta(k^2 \epsilon).$$

Once the latter is established, the proof of Lemma 3.7 follows directly by weak duality. In particular,

$$\sum_s \mu(s) \cdot r(s) \geq Z_{LP}^\star \geq \mu - \epsilon \cdot \sum_{i \in [n]} \sum_{s_i \in \mathcal{S}_i} \sum_{s_i' \in \mathcal{S}_i} \lambda_{s_i s_i'}^i$$
$$\geq v_2 - 3/k - \Theta(k^2 \epsilon).$$

### 3.1. The Dual-Fitting Argument

In this section, we present our assignment of the dual variables $\{\mu, \lambda_{s_i, s_{-i}}^i\}$. We use the notation $\hat{\lambda}_{s_i, s_{-i}}^i$ to differentiate between the assignment and the variables $\lambda_{s_i, s_{-i}}^i$. We also remind that without loss of generality we have assumed that $v_1 \geq v_2 \geq \ldots \geq v_n$. In Definition 3.5 we present the assignment of the dual variables corresponding to any bidder $i \geq 3$.

**Definition 3.5.** *For each agent $i \geq 3$, $\hat{\lambda}_{s_i s_i'}^i := 0$.*

In Definition 3.6 we present the assignment of the dual variables $\hat{\lambda}_{s_i s_i'}^i$ for the bidders $i \in \{1, 2\}$.

**Definition 3.6.** *Let a bidder $i \in \{1, 2\}$. Then for all bids $s_i \in \mathcal{S}_i$,*

$$\hat{\lambda}_{s_i s_i'}^i := \begin{cases} 0 & \text{if } s_i' \leq s_i \\ 1 & \text{if } s_i + 1/k \leq s_i' \leq v_2 - 2/k \\ 0 & \text{if } v_2 - 1/k \leq s_i' \end{cases}$$

To simplify notation we consider the following notation,

$$\hat{b}_s := r(s) - \sum_{i \in [n]} \sum_{s_i' \in \mathcal{S}_i} \hat{\lambda}^i_{s_i s_i'} \cdot (U_i(s_i, s_{-i}) - U_i(s_i', s_{-i})).$$

In Lemma 3.7 we state the main technical result of the section, stating a lower bound on each $\hat{b}_s$.

**Lemma 3.7.** *Let the assignment $\hat{\lambda}^i_{s_i s_i'}$ described in Definition 3.5 and 3.6 respectively. Then for all bidding profiles $s \in \mathcal{S}$, $\hat{b}_s \geq 1 - 3/k$.*

The proof of Lemma 3.7 is presented at the end of this section. Up next, we state and prove Lemma 3.8 that directly implies Theorem 2.6.

**Lemma 3.8.** *Let the assignment $\hat{\lambda}^i_{s_i s_i'}$ of Definition 3.5,3.6 and $\hat{\mu} := v_2 - 3/k$. This assignment is dual feasible and additionally*

$$\hat{\mu} - \epsilon \cdot \sum_{i \in [n]} \sum_{s_i \in \mathcal{S}_i} \sum_{s_i' \in \mathcal{S}_i} \hat{\lambda}^i_{s_i s_i'} \geq v_2 - 3/k - \Theta(k^2 \epsilon).$$

*Proof of Lemma 3.8.* Since $\hat{\mu} = v_2 - 3/k$ then by Lemma 3.7 we directly get that for all $s \in \mathcal{S}$,

$$\hat{\mu} + \sum_{i \in [n]} \sum_{s_i' \in \mathcal{S}_i} \hat{\lambda}^i_{s_i s_i'} \cdot (U_i(s_i, s_{-i}) - U_i(s_i', s_{-i})) \leq r(s)$$

which means that the assignment is dual feasible. By Definition 3.5 and 3.6 we get that $\hat{\mu} - \epsilon \sum_{i \in [n]} \sum_{s_i, s_i' \in \mathcal{S}_i} \hat{\lambda}^i_{s_i s_i'} :=$

$$v_2 - \frac{3}{k} - \epsilon \sum_{i=1}^{2} \sum_{s_i \in \mathcal{S}_i} \sum_{s_i' = s_i + 1/k}^{v_2 - 2/k} 1 \geq v_2 - \frac{3}{k} - 2\epsilon \cdot k^2$$

$$\square$$

We conclude the section with the proof of Lemma 3.7.

*Proof of Lemma 3.7.* By Definition 3.5 we know that for all bidders $i \geq 3$, $\hat{\lambda}^i_{s_i s_i'} = 0$. Thus for all $s \in \mathcal{S}$,

$$\hat{b}_s := r(s) - \sum_{i \in \{1,2\}} \sum_{s_i' \in \mathcal{S}_i} \hat{\lambda}^i_{s_i s_i'} (U_i(s_i, s_{-i}) - U_i(s_i', s_{-i}))$$

Notice that by Definition 3.6 we get that

$$\hat{b}_s := r(s) - \sum_{i \in \{1,2\}} \sum_{s_i' = s_i + 1/k}^{v_2 - 2/k} (U_i(s_i, s_{-i}) - U_i(s_i', s_{-i})).$$

$$(2)$$

We complete the proof by partitioning $\mathcal{S}$ in the following 4 classes and establishing that $\hat{b}_s \geq 1 - 3/k$ for each one of them separately.

**Lemma 3.9.** *Let a bidding profile $s \in \mathcal{S}$ such that $s_1 = \max(s_1, \ldots, s_n)$ and $s_2 < s_1$ then $b_s \geq v_2 - 2/k$.*

**Lemma 3.10.** *Let a bidding profile $s \in \mathcal{S}$ such that $s_2 = \max(s_1, \ldots, s_n)$ and $s_1 < s_2$ then $b_s \geq v_2 - 2/k$.*

**Lemma 3.11.** *Let a bidding profile $s \in \mathcal{S}$ such that $s_2 = \max(s_1, \ldots, s_n)$ and $s_1 = s_2$ then $b_s \geq v_1 - 2/k$.*

**Lemma 3.12.** *Let a bidding profile $s \in \mathcal{S}$ such that $s_1 \neq \max(s_1, \ldots, s_n)$ and $s_2 \neq \max(s_1, \ldots, s_n)$[4] then $b_s \geq v_2 - 3/k$.*

The proof of Lemma 3.9 is presented in Section 3.2. The proof of Lemma 3.10 is similar and is deferred in Appendix D. Due to lack of space, the proofs of Lemma 3.11 and Lemma 3.12 are also deferred to Appendix D. $\square$

### 3.2. Proof of Lemma 3.9

In this section we provide the proof of Lemma 3.9.

**Lemma 3.13.** *Let a bidding profile $s \in \mathcal{S}$ such that $s_1 = \max(s_1, \ldots, s_n)$ and $s_2 < s_1$ then $b_s \geq v_2 - 2/k$.*

*Proof.* Since $s_1 = \max(s_1, \ldots, s_n)$ and $s_2 < s_1$, we know that bidder 2 does not win the item. Thus $U_2(s_2, s_{-2}) = 0$. Since $s_1 = \max(s_1, \ldots, s_n)$ we know that bidder 1 is among the winners meaning that $U_1(s_1, s_{-1}) \leq v_1 - s_1$. Notice that $U_1(s_1, s_{-1})$ does not necessarily equals $v_1 - s_1$ since in case of a tie, the item is given to one of the winner uniformly at random.

Since $s_1 = \max(s_1, \ldots, s_n)$, we also know that if bidder 1 submits any bid $s_1' \geq s_1 + \frac{1}{k}$, it will be the only winner and thus $U_1(s_1', s_{-1}) = v_1 - s_1'$. By Equation 2 in the proof of Lemma 3.7 we know that

$$\hat{b}_s = s_1 - \underbrace{\sum_{s_1' = s_1 + 1/k}^{v_2 - 2/k} (U_1(s_1, s_{-1}) - U_1(s_1', s_{-1}))}_{(A)}$$

$$- \underbrace{\sum_{s_2' = s_2 + 1/k}^{v_2 - 2/k} (U_2(s_2, s_{-2}) - U_2(s_2', s_{-2}))}_{(B)}$$

Up next, we upper bound the terms (A) and (B) separately. Starting with the term (A), we first remind that

---

[4]Notice that the fact that $v_1 \geq v_2 \geq \ldots \geq v_n$ does imply that the bids $s_1, s_2, \ldots, s_n$ follow the same order.

$U_1(s_1', s_{-1}) = v_1 - s_1'$ and $U_1(s_1, s_{-1}) \leq v_1 - s_1$. Thus,

$$
\begin{aligned}
(A) \quad &:= \quad \sum_{s_1'=s_1+1/k}^{v_2-2/k} (U_1(s_1, s_{-1}) - U_1(s_1', s_{-1})) \\
&\leq \quad \sum_{s_1'=s_1+1/k}^{v_2-2/k} ((v_1 - s_1) - (v_1 - s_1')) \\
&= \quad \sum_{s_1'=s_1+1/k}^{v_2-2/k} (s_1' - s_1) \\
&= \quad \frac{1}{k}\left(\frac{(kv_2-1)(kv_2-2)}{2} - \frac{ks_1(ks_1+1)}{2}\right) \\
&\quad - \; s_1 \cdot (kv_2 - 2 - k \cdot s_1 - 1 + 1) \\
&= \quad \frac{1}{2k}\left(k^2 \cdot v_2^2 - 3k \cdot v_2 + 2 - k^2 \cdot s_1^2 - k \cdot s_1\right) \\
&\quad - \; \frac{1}{k}\left(k^2 \cdot s_1 v_2 - 2k \cdot s_1 - k^2 \cdot s_1^2\right)
\end{aligned}
$$

We now continue with (B). First notice that bidder 2 does not win the item since $s_2 < s_1$. Thus, $U_2(s_2, s_{-2}) = 0$. For the exact same reason $U_2(s_2', s_{-2}) = 0$ for all bids $s_2' \leq s_1 - 1$. As a result,

$$
\begin{aligned}
(B) \quad &:= \quad \sum_{s_2'=s_2+1/k}^{v_2-2/k} (U_2(s_2, s_{-2}) - U_2(s_2', s_{-2})) \\
&= \quad \sum_{s_2'=s_2+1/k}^{s_1-1} \left(\underbrace{U_2(s_2, s_{-2})}_{0} - \underbrace{U_2(s_2', s_{-2})}_{0}\right) \\
&\quad + \quad \sum_{s_2'=s_1}^{v_2-1/k} \left(\underbrace{U_2(s_2, s_{-2})}_{0} - U_2(s_2', s_{-2})\right) \\
&= \quad -U_2(s_1, s_{-2}) - \sum_{s_2'=s_1+1/k}^{v_2-2/k} \underbrace{U_2(s_2', s_{-i})}_{v_2 - s_2'} \\
&\qquad \text{since agent 2 wins with } s_2' \geq s_1 + 1/k \\
&= \quad -\underbrace{U_2(s_1, s_{-2})}_{\leq 0} - \sum_{s_2'=s_1+1/k}^{v_2-2/k} (v_2 - s_2') \\
&\leq \quad -v_2(k \cdot v_2 - 2 - k \cdot s_1 - 1 + 1) \\
&\quad + \; \frac{1}{k}\left(\frac{(kv_2-1)(kv_2-2)}{2} - \frac{ks_1(ks_1+1)}{2}\right) \\
&\leq \quad -\frac{1}{k}(k^2 \cdot v_2^2 - 2v_2 k - k^2 \cdot s_1 v_2) \\
&\quad + \; \frac{1}{2k}\left(k^2 \cdot v_2^2 - 3k \cdot v_2 + 2 - k^2 \cdot s_1^2 - k \cdot s_1\right)
\end{aligned}
$$

Putting everything together we get that,

$$
\begin{aligned}
(A) + (B) \quad &\leq \quad \frac{1}{k}\left(k^2 \cdot v_2^2 - 3k \cdot v_2 + 2 - k^2 \cdot s_1^2 - k \cdot s_1\right) \\
&\quad - \; \frac{1}{k}\left(k^2 \cdot s_1 v_2 - 2k \cdot s_1 - k^2 \cdot s_1^2\right) \\
&\quad - \; \frac{1}{k}(k^2 \cdot v_2^2 - 2k \cdot v_2 - k^2 \cdot s_1 v_2) \\
&= \quad -v_2 + \frac{2}{k} + s_1
\end{aligned}
$$

Thus, $\hat{b}_s := s_1 - (A) - (B) \geq v_2 - \frac{2}{k}$. $\qquad \square$

## 4. Experimental Evaluations

In this section, we experimentally evaluate the time-average revenue produced when all bidders use the no-swap regret algorithm of (Blum & Mansour, 2007), which guarantees $\mathcal{O}(\sqrt{kT})$ swap regret.

As discussed earlier, Theorem 2.6 implies that, with high probability,

$$
\frac{1}{T}\sum_{t=1}^{T} r(t) \geq v_2 - \frac{3}{k} - \mathcal{O}\left(\frac{k^{2.5}}{\sqrt{T}}\right). \tag{3}
$$

We first consider a discrete first-price auction with $k = 10$ and 3 bidders with valuations $v_1 = 1 \geq v_2 \geq v_3 = 0$. To capture different cases, we examine $v_2 \in \{0, 0.2, 0.5, 0.8, 1\}$. Figure 1 presents our results, showing that in all cases, the time-average revenue always surpasses $v_2 - 1/k = v_2 - 0.1$, significantly faster than $\mathcal{O}(k^7)$.

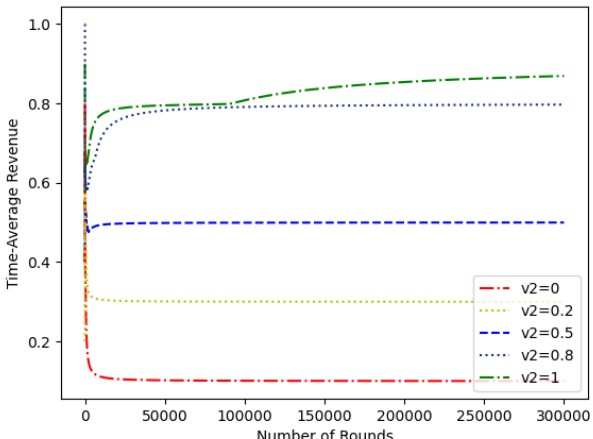

*Figure 1.* Time-averaged revenue for different values of $v_2$.

Next, we examine the trade-off between the discretization level $k$ and produced revenue. Equation (3) suggests that larger $k$ values yield higher long-term revenue but at the cost of slower convergence. To evaluate this, we consider 3 bidders with valuations $v_1 = 1$, $v_2 = 1$, and $v_3 = 0$, participating in auctions with $k \in \{10, 30, 50\}$. Figure 2 confirms that higher $k$ values indeed lead to greater revenue but a slower rate. Additionally, our second experiment confirms that revenue converges to $v_2$ much faster than our theoretical analysis predicts.

We now experimentally evaluate the revenue produced by no-swap regret dynamics in the Bayesian setting where the valuation of each bidder $i \in [n]$ is sampled according to a probability distribution $\mathcal{P}_i \in \Delta(\mathcal{B})$. We consider the i.i.d.

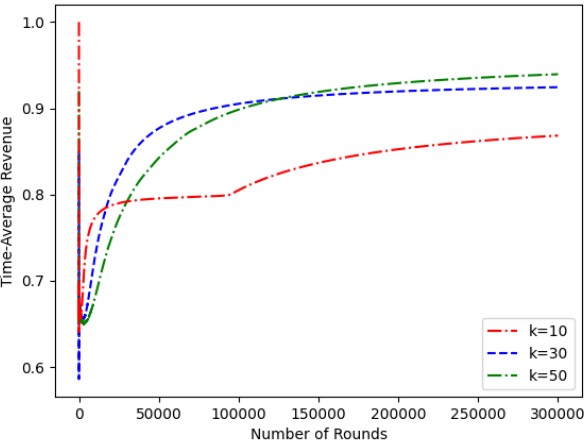

*Figure 2.* Impact of discretization level $k$ on revenue convergence.

setting where each bidder $i \in [n]$ admits either $v_i = 0$ or $v_i = 1$ with probability $1/2$. In case of $n$ bidders then expected second highest valuation admits $\mathbb{E}[v_2] = 1 - (n + 1)/2^{-n}$. We evaluate the algorithm of (Blum & Mansour, 2007) for $n = \{2, 3, 4, 5\}$ and $k = 10$. We remark that in the Bayesian setting each bidder runs on parallel two different no-swap regret algorithms - one responsible for the bids in case $v_i = 0$ and one responsible for the bids in case $v_i = 1$. In the following figure, we present our experimental findings indicating that even in the Bayesian setting the revenue produced converges to $\mathbb{E}[v_2]$.

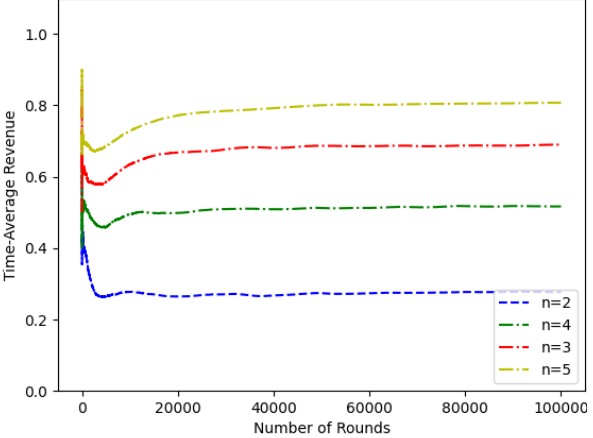

*Figure 3.* Revenue produced for different number of bidders.

We additionally examine the convergence properties of no-swap regret dynamics to Bayes Mixed Nash Equilibrium for $n = 2$ and i.i.d. valuations uniform in $\{0, 1\}$ (exactly as above). In case of continuous bidding space $\mathcal{B} = [0, 1]$, there exists a unique Bayes Mixed Nash Equilibrium $s^\star$ where $s^\star(0) = 0$ and $s^\star(1) \sim \mathcal{P}$ with CDF, $F_{\mathcal{P}}(x) = 1/(1-x) - 1$ for all $x \in [0, 1/2]$. Our experimental findings reveal that the time-average bidding behavior of the bidders

is very close to Bayes Mixed Nash Equilibrium. In particular for $v_i = 0$, bidder $i$ always bids $0$ while for $v_i = 1$ bidder $i$ randomly selects its bid according to a CDF that closely approximates $F_{\mathcal{P}}(x) = \frac{1}{1-x} - 1$ for $x \in [0, 1/2]$.

We have deferred figures and a more comprehensive explanation of our experiments in the Bayesian Setting to the Appendix E, but our results complement the experimental findings of (Bichler et al., 2023) (see Remark E.1).

## 5. Conclusion and Future Directions

In this work, we study the revenue guarantees of approximate correlated equilibrium in discrete first-price auctions. In particular, we establish that any $\epsilon$-approximate correlated equilibrium guarantees revenue of at least $v_2 - \Theta(1/k) - \Theta(k^2 \cdot \epsilon)$. Our results imply that if all bidders use no-swap regret algorithms, then the time-averaged revenue converges close to the second-highest valuation within a polynomial number of rounds with respect to the number of possible bids. To the best of our knowledge, this is the first result quantifying the convergence properties of the revenue produced by online learning agents in first-price auctions.

Our work leaves open several interesting research directions. First, achieving faster convergence rates for revenue is a compelling direction for future research. Our work leaves a gap between the upper and lower bounds of an $\epsilon$-approximate correlated equilibrium. Specifically, Theorem 2.6 establishes a bound of $v_2 - \Theta(1/k) - \Theta(k^2 \cdot \epsilon)$, whereas Theorem 2.8 provides a bound of $v_2 - \Theta(1/k) - \Theta(\epsilon)$. Since our experimental evaluations indicate faster convergence rates than the ones predicted by Theorem 2.6, we conjecture that the right bound is the one given in Theorem 2.8. Another very interesting direction is extending our dual fitting approach in the Bayesian setting, where bidders' valuations are i.i.d. Our experimental results indicate that the revenue generated by no-swap regret agents indeed approaches the expected value of the second-highest valuation.

**Acknowledgments** This project was supported by the Villum Young Investigator Award (Grant no. 72091).

## Impact Statement

This paper presents work whose goal is to advance the field of Machine Learning. There are many potential societal consequences of our work, none which we feel must be specifically highlighted here.

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

# A. Revisiting the argument in (Feldman et al., 2016)

In this section we establish Theorem 2.5 by extending the very elegant argument of (Feldman et al., 2016) in the context of discrete first-price auctions. We note that the purpose of this section is solely to provide intuition and the reader can directly proceed to Section 3 where the proof of Theorem 2.6 is presented.

**Theorem 2.5.** *Let the probability distribution $\mu \in \Delta(\mathcal{S})$ be an exact correlated equilibrium ($\epsilon = 0$). Then its expected revenue admits $\sum_{s \in \mathcal{S}} \mu(s) \cdot r(s) \geq v_2 - 4/k$.*

*Proof.* To simplify notation we denote $\Pr_\mu[\cdot]$ as $\Pr[\cdot]$. Let $p^\star \in \mathcal{B}$ denote the minimum bid with non-zero probability that can win the item,

$$p^\star := \min\{p \in \mathcal{B} : \Pr[\max_{k \in [n]} s_k \leq p] > 0\}.$$

In case $p^\star \geq v_2 - 3/k$ notice that the proof is already completed. Thus without loss of generality we assume that $p^\star \leq v_2 - 4/k$. Thus there exists a bid $p \in \mathcal{B}$ such that $p \in [p^\star, \frac{p^\star + v_2}{2} - \frac{1}{k})$.

Now consider the bidders $\{1, 2\}$ and notice that

$$\sum_{j \in \{1,2\}} \Pr[s_j = \max_{k \in [n]} s_k \text{ and } s_j \leq p] \leq \Pr[\max_{k \in [n]} s_k \leq p].$$

Thus, there exists a bidder $j \in \{1, 2\}$ such that

$$\Pr[s_j = \max_{k \in [n]} s_k \text{ and } s_j \leq p] \leq \frac{1}{2} \cdot \Pr[\max_{k \in [n]} s_k \leq p.] \tag{4}$$

Up next, we construct the following switching function for this bidder $j \in \{1, 2\}$, $\delta : \mathcal{S}_j \mapsto \mathcal{S}_j$,

$$\delta(s_j) = \begin{cases} p + 1/k & \text{if } s_j \leq p \\ s_j & \text{otherwise} \end{cases}.$$

Since $\mu \in \Delta(\mathcal{S})$ is an exact correlated equilibrium we know that,

$$\mathbb{E}[U_j(\delta(s_j), s_{-j})] - \mathbb{E}[U_j(s_j, s_{-j})] \leq 0. \tag{5}$$

Let $\mathcal{S}' := \{s \in \mathcal{S} : s_j \leq p\}$. Notice that for any $s \notin \mathcal{S}'$, $\delta(s_j) = s_j$. As a result,

$$\mathbb{E}[U_j(\delta(s_j), s_{-j})] - \mathbb{E}[U_j(s_j, s_{-j})] = \sum_{s \in \mathcal{S}'} \mu(s) \cdot U_j(\delta(s_j), s_{-j}) - \sum_{s \in \mathcal{S}'} \mu(s) \cdot U_j(s_j, s_{-j}). \tag{6}$$

Let us further partition $\mathcal{S}' := (\mathcal{S}', \mathcal{S}'/\mathcal{S}'')$ where $\mathcal{S}'' := \{s \in \mathcal{S}' : s_j = \max_{k \in [n]} s_k\}$. In simpler words, $\mathcal{S}''$ contains the bidding profiles of $s \in \mathcal{S}'$ at which agent $j \in \{1, 2\}$ wins the item. Notice that for any $s \notin \mathcal{S}''$, $U_j(s_j, s_{-j}) = 0$ and $U_j(\delta(s_j), s_{-j}) = v_j - s_j$ since $s_j$ is not the maximum bid while $\delta(s_j) = p + 1/k$ is definitely the maximum bid[5]. As a result,

$$\sum_{s \notin \mathcal{S}''} \mu(s) \cdot (U_j(\delta(s_j), s_{-j}) - U_j(s_j, s_{-j})) = (v_2 - p - 1/k) \cdot \Pr[s_j \neq \max_{k \in [n]} s_k \text{ and } s_j \leq p]$$

$$\geq (v_2 - p - 1/k) \cdot \Pr[\max_{k \in [n]} s_k \leq p] \cdot \frac{1}{2} \tag{7}$$

Notice that for any $s \in \mathcal{S}''$, $U_j(s_j, s_{-j}) \leq v_j - s_j$ and $U_j(\delta(s_j), s_{-j}) = v_j - \delta(s_j)$. As a result,

$$\sum_{s \in \mathcal{S}''} \mu(s) \cdot (U_j(\delta(s_j), s_{-j}) - U_j(s_j, s_{-j})) \geq \sum_{s \in \mathcal{S}''} \mu(s) \cdot (s_j - \delta(s_j))$$

$$\geq (p^\star - p - 1/k) \Pr[s_j = \max_{k \in [n]} s_k \text{ and } s_j \leq p] \tag{8}$$

$$\geq (p^\star - p - 1/k) \Pr[s_j = \max_{k \in [n]} s_k]/2 \tag{9}$$

---

[5] $\mathcal{S}'' \subseteq \mathcal{S}'$ and thus $\max_{k \in [n]} s_k \leq p$

where Equation (15) is due to $p^\star := \min\{p \in \mathcal{B} : \Pr[\max_{k \in [n]} s_k \leq p] > 0\}$. Then by Equation (7) and (9) we get that

$$\Pr[\max_{k \in [n]} s_k \leq p] \cdot \left( \frac{v_2 + p^\star}{2} - p - 1/k \right) \leq \sum_{s \notin \mathcal{S}'} \mu(s) \cdot (U_j(\delta(s_j), s_{-j}) - U_j(s_j, s_{-j})) \leq 0$$

which is a contradiction since $p \leq \frac{v_2 + p^\star}{2} - 1/k$.  $\square$

In the rest of the section, we sketch how the arguments in the proof of Theorem 2.5 can be extended so as to establish that an $\epsilon$-approximate correlated equilibrium admits revenue greater than $v_2 - \mathcal{O}(1/k) - \epsilon \cdot k^{\Theta(\log k)}$. The latter approach is presented in further detail in Appendix A.1.

Up next we build upon the proof of Theorem 2.5. Let us assume that $\mu$ is an $\epsilon$-approximate correlated equilibrium. Then by selecting $p^\star = 0$ and using in the RHS of Equation (5), $\epsilon > 0$ instead of $0$, one could establish that for all $p \in \mathcal{B}$ with $p \in \left[0, \frac{v_2}{2} - \frac{1}{k}\right)$,

$$\Pr[\max_{k \in [n]} s_k \leq p] \left( \frac{v_2 + p^\star}{2} - p - 1/k \right) \leq \epsilon. \tag{10}$$

As a result, one could conclude that

$$\Pr[\max_{k \in [n]} s_k \leq p] \leq \mathcal{O}(\epsilon k) \text{ for all } p \in \left[0, \frac{v_2}{2} - \frac{1}{k}\right). \tag{11}$$

Then the exact same arguments can be repeated for $p^\star = v_2/2$ and one can conclude that

$$\Pr[\max_{k \in [n]} s_k \leq p] \leq \mathcal{O}(\epsilon k^2) \text{ for all } p \in \left[\frac{v_2}{2}, \frac{3v_2}{4} - \frac{1}{k}\right). \tag{12}$$

The extra $k$ in Equation (12), comes from the fact that the probability of agent $k$ winning the item with bid $s_k \leq p^\star - 1$ is not $0$ (as in the proof of Theorem 2.5) but rather $\mathcal{O}(k\epsilon)$ (see Equation (11)). Then by the fact that $\left( \frac{v_2 + p^\star}{2} - p - 1/k \right) \geq 1/k$ we get the $\mathcal{O}(\epsilon k^2)$ bound (see Claim A.1).

By repeating the same argument inductively, one can argue that for all $\alpha \leq \log(k/3) - 1$,

$$\Pr[\max_{k \in [n]} s_k \leq p] \leq \mathcal{O}(\epsilon \cdot k^a) \text{ for all } p \in \Delta(a)$$

where $\Delta(a) := \left[ (1 - 2^{-a})v_2, (1 - 2^{-a-1})v_2 - \frac{1}{k} \right]$ (see Claim A.1). By using the latter, one can argue that the revenue of an $\epsilon$-correlated equilibrium is at least $v_2 - \frac{1}{k} - \epsilon \cdot k^{\mathcal{O}(\log k)}$ (see Claim A.3).

### A.1. Omitted Proofs of Section A

In this section we present a more detailed sketch on how the argument of (Feldman et al., 2016) presented in Appendix A can be extended to $\epsilon$-approximate correlated equilibrium. We remind that the goal is to establish that any $\epsilon$-approximate correlated equilibrium $\mu(\cdot)$, its revenue $\sum_s \mu(s) \cdot r(s) \geq v_2 - \mathcal{O}(1/k) - k^{\mathcal{O}(\log k)} \cdot \epsilon$.

We start by first upper bounding the probability the revenue to be less than a value $p$, $\Pr[\max(s_1, \ldots, s_n) \leq p]$.

**Claim A.1.** *Let an integer* $\alpha \leq \log(k/3) - 1$. *Then for any* $p \in \left[ (1 - 2^{-a})v_2, (1 - 2^{-a-1})v_2 - \frac{1}{k} \right)$

$$\Pr[\max(s_1, \ldots, s_n) \leq p] \leq \mathcal{O}(\epsilon \cdot k^{a+1})$$

*Proof Sketch.* The case of $a = 0$ can be established with the argument presented in Section A.

Up next, we present the induction Step. Let $p^\star = (1 - 2^{-a-1})v_2$. By the induction hypothesis we know that $\Pr[\max(s_1, \ldots, s_n) \leq p^\star - 1/k] \leq \mathcal{O}(\epsilon \cdot k^{a+1})$. Since $a \leq \log(k/3) - 1$ there exists at least one valid bid $p \in \left[ p^\star, \frac{p^\star + v_2}{2} - \frac{1}{k} \right)$. As in the proof of Theorem 2.5, there exists a bidder $j \in \{1, 2\}$ such that

$$\Pr\left[ s_j = \max_{k \in [n]} s_k \text{ and } s_j \leq p \right] \leq \frac{1}{2} \cdot \Pr\left[ \max_{k \in [n]} s_k \right]$$

Now construct the following switching function $\delta(\cdot)$ for agent $j$,

$$\delta(s_j) = \begin{cases} p + 1/k & \text{if } s_j \leq p \\ s_j & \text{otherwise} \end{cases}$$

To simplify notation let $A := \mathbb{E}[U_j(\delta(s_j), s_{-j})] - \mathbb{E}[s_j, s_{-j})] \leq \epsilon$. Since $\mu$ is an $\epsilon$-approximate correlated equilibrium, we know that $A \leq \epsilon$. Up next we present Proposition D.6 that provides a lower bound on $A$.

**Proposition A.2.** $A \geq \Pr[\max_{k \in [n]} s_k \leq p] \cdot \left( \frac{v_2 + p^\star}{2} - p - 1/k \right) - \Pr[\max_{k \in [n]} s_k \leq p^\star - 1]$

*Proof.* Let $\mathcal{S}' := \{s \in \mathcal{S} : s_k \leq p\}$. Notice that for any $s \notin \mathcal{S}'$, $\delta(s_j) = s_j$. As a result,

$$A = \sum_{s \in \mathcal{S}'} \mu(s) \cdot U_j(\delta(s_j), s_{-j}) - \sum_{s \in \mathcal{S}'} \mu(s) \cdot U_j(s_j, s_{-j}). \tag{13}$$

Let us further partition $\mathcal{S}' := (\mathcal{S}', \mathcal{S}'/\mathcal{S}'')$ where $\mathcal{S}'' := \{s \in \mathcal{S}' : s_j = \max_{k \in [n]} s_k\}$. In simpler words, $\mathcal{S}''$ contains the bidding profiles of $s \in \mathcal{S}'$ at which agent $j \in \{1, 2\}$ wins the item. Notice that for any $s \notin \mathcal{S}''$, $U_j(s_j, s_{-j}) = 0$ and $U_j(\delta(s_j), s_{-j}) = v_j - s_j$ since $s_j$ is not the maximum bid while $\delta(s_j) = p + 1/k$ is definitely the maximum bid[6]. As a result,

$$\sum_{s \notin \mathcal{S}''} \mu(s) \cdot (U_j(\delta(s_j), s_{-j}) - U_j(s_j, s_{-j})) \geq (v_2 - p - 1/k) \cdot \Pr[s_j \neq \max_{k \in [n]} s_k \text{ and } s_j \leq p]$$

$$\geq (v_2 - p - 1/k) \cdot \Pr[\max_{k \in [n]} s_k \leq p] \cdot \frac{1}{2} \tag{14}$$

where the last inequality comes from the fact that $\Pr\left[s_j = \max_{k \in [n]} s_k \text{ and } s_j \leq p\right] \leq \frac{1}{2} \cdot \Pr\left[\max_{k \in [n]} s_k\right]$. Notice that for any $s \in \mathcal{S}''$, $U_j(s_j, s_{-j}) \leq v_j - s_j$ and $U_j(\delta(s_j), s_{-j}) = v_j - \delta(s_j)$. As a result,

$$\sum_{s \in \mathcal{S}''} \mu(s) \cdot (U_j(\delta(s_j), s_{-j}) - U_j(s_j, s_{-j})) \geq \sum_{s \in \mathcal{S}''} \mu(s) \cdot (s_j - \delta(s_j))$$

$$\geq (p^\star - p - 1/k) \Pr[s_j = \max_{k \in [n]} s_k \text{ and } p^\star \leq s_j \leq p]$$

$$+ (0 - p - 1/k) \Pr[s_j = \max_{k \in [n]} s_j \text{ and } s_j \leq p^\star - 1]$$

$$\geq (p^\star - p - 1/k) \Pr[\max_{k \in [n]} s_k \leq p]/2$$

$$- \Pr[\max_{k \in [n]} s_k \leq p^\star - 1] \tag{15}$$

Then by adding Equation (14) and (15) we get that

$$\Pr[\max_{k \in [n]} s_k \leq p] \cdot \left( \frac{v_2 + p^\star}{2} - p - 1/k \right) - \Pr[\max_{k \in [n]} s_k \leq p^\star - 1] \leq A$$

$\square$

Now by combining Proposition D.6 with the fact that $A \leq \epsilon$ we get that,

$$\Pr\left[\max(s_1, \ldots, s_n) \leq p\right] \cdot \left( \frac{v_2 + p^\star}{2} - p - 1/k \right) \leq \Pr\left[\max(s_1, \ldots, s_n) \leq p^\star - 1\right] + \epsilon$$

Since $\left( \frac{v_2 + p^\star}{2} - p - 1/k \right) \geq 1/k$ we finally get that

$$\Pr\left[\max(s_1, \ldots, s_n) \leq p\right] \leq k \cdot \Pr\left[\max(s_1, \ldots, s_n) \leq p^\star - 1\right] + \epsilon \cdot k \leq \mathcal{O}(k^{a+2} \cdot \epsilon)$$

which completes the induction step. $\square$

---

[6] $\mathcal{S}'' \subseteq \mathcal{S}'$ and thus $\max_{k \in [n]} s_k \leq p$

**Claim A.3.** *Let an approximate $\epsilon$-approximate correlated equilibrium $\mu(\cdot)$ then $\sum_s \mu(s) \cdot r_s \geq v_2 - \mathcal{O}(1/k) - k^{\mathcal{O}(\log k)} \epsilon$.*

*Proof Sketch.*

$$
\begin{aligned}
\sum_s \mu(s) \cdot r_s &= \sum_{p=0}^{1} \Pr[\max(s_1, \ldots, s_n) \geq p]/k \\
&\geq \sum_{p=0}^{v_2(1-3/k)} \Pr[\max(s_1, \ldots, s_n) \geq p]/k \\
&= \sum_{p=0}^{v_2(1-3/k)} \left(1 - \Pr[\max(s_1, \ldots, s_n) \leq p]\right)/k \\
&= v_2(1 - 3/k) - \frac{1}{k} \sum_{a=0}^{\log(k/3)} \sum_{p=(1-2^{-a})v_2}^{(1-2^{-a-1})v_2 - 1/k} \mathcal{O}(\epsilon \cdot k^{\alpha+1}) \\
&\geq v_2(1 - 3/k) - \sum_{a=0}^{\log(k/3)-1} \frac{2^{-a}}{2} \cdot \mathcal{O}(k^a \cdot \epsilon) \\
&\geq v_2 - \mathcal{O}(1/k) - k^{\mathcal{O}(\log k)} \cdot \epsilon
\end{aligned}
$$

$\square$

## B. Proof of Theorem 2.8

**Theorem 2.8.** *Let the discrete-price auctions with 2 bidders with $v_1 = v_2 = 1$. There exists an $\epsilon$-approximate correlated equilibrium $\mu \in \Delta(\mathcal{S})$ such that the expected payoff admits*

$$
\sum_{s \in \mathcal{S}} \mu(s) \cdot r(s) \leq 1 - \Theta(1/k) - \Theta(\epsilon)
$$

*Proof.* Let us consider the following joint distribution $\mu \in \Delta(\mathcal{S})$,

$$
\mu(s_1, s_2) = \begin{cases} 0 & \text{if } s_1 \neq s_2 \\ \epsilon/k & \text{if } s_1 = s_2 \leq 1 - 1/k \\ 1 - \epsilon & \text{if } s_1 = s_2 = 1 \end{cases}
$$

We first show that $\mu \in \Delta(\mathcal{S})$ is an $\epsilon$-approximate correlated equilibrium. Let bidder $i \in \{1, 2\}$, then for any $s = (s_i, s_{-i})$ with $s_i \leq 1 - 1/k$, the best response would be bidding $s_i + 1/k$ since in this case it would win the item for sure instead of probability $1/2$. As a result, for any switching function $\delta : \mathcal{S}_i \mapsto \mathcal{S}_i$,

$$
\begin{aligned}
\mathbb{E}_{s \sim \mu}[U_i(\delta(s_i), s_{-i})] - \mathbb{E}_{s \sim \mu}[U_i(s_i, s_{-i})] &\leq \sum_{j=0}^{1-1/k} \left(1 - \frac{j+1}{k}\right) \cdot \frac{\epsilon}{k} - \sum_{j=0}^{1-1/k} \left(1 - \frac{j}{k}\right) \cdot \frac{\epsilon}{2k} \\
&= \sum_{j=0}^{1-1/k} \left(\frac{1}{2} - \frac{1}{k}\right) \cdot \frac{\epsilon}{k} \leq \epsilon
\end{aligned}
\tag{16}
$$

Now concerning the revenue of the this correlated equilibrium $\mu \in \Delta(S)$,

$$
\sum_{s \in \mathcal{S}} \mu(s) \cdot r(s) = 1 - \epsilon + \frac{\epsilon}{k} \cdot \sum_{j=1}^{1-1/k} j = 1 - \epsilon + \frac{\epsilon}{k^2} \cdot \frac{(k-1)k}{2} = 1 - \frac{\epsilon}{2} - \frac{\epsilon}{2k}
$$

$\square$

## C. Proof of Theorem 2.3

**Theorem 2.3.** *If each bidder $i \in [n]$, selects its bid $s_i^t \in \mathcal{S}_i$ according to a no-swap regret algorithm $\mathcal{A}_i$, $s_i^t \sim x_i^t$. Then after $T = \mathcal{O}\left(\max_{i \in [n]} \mathcal{R}_i(T)/\epsilon + \log(n/\delta)/\epsilon\right)$ rounds, the time-average distribution $\hat{\mu} \in \Delta(\mathcal{S})$, defined as*

$$\hat{\mu}(s) := \frac{1}{T} \sum_{t=1}^{T} \mathrm{I}[s_t = s] \quad \text{for all } s \in \mathcal{S},$$

*is an $\epsilon$-approx. correlated equilibrium with probability $1 - \delta$.*

*Proof.* We remind that by Chernoff bounds, if $X_1, ..., X_n$ are independent random variables taking values in $[0, 1]$. Then for any $\delta > 0$,

$$\Pr\left[\sum_{t=1}^{T} X_t \geq (1+\delta) \sum_{t=1}^{T} \mathbb{E}\left[X_t\right]\right] \leq e^{-\delta \cdot \sum_{t=1}^{T} \mathbb{E}[X_t]} \tag{17}$$

Let us fix an agent $i \in [n]$ as well as a switching function $\delta : \mathcal{S}_i \to \mathcal{S}_i$. Now consider $X_t := U_i(\delta(s_i^t), s_{-i}^t) - U_i(s_i^t, s_{-i}^t)$ and notice that $\sum_{t=1}^{T} \mathbb{E}\left[X_t\right] \leq \mathcal{R}_i^{\mathrm{swap}}(T)$. Then $\delta = \epsilon T / \sum_{t=1}^{T} \mathbb{E}\left[X_t\right] - 1$ and Equation (17) we get that,

$$\Pr\left[\sum_{t=1}^{T} X_t \geq \epsilon \cdot T\right] = \Pr\left[\sum_{t=1}^{T} X_t \geq (1+\delta) \sum_{t=1}^{T} \mathbb{E}\left[X_t\right]\right] \leq e^{-\left(\epsilon T / \sum_{t=1}^{T} \mathbb{E}[X_t] - 1\right) \cdot \sum_{t=1}^{T} \mathbb{E}[X_t]} \leq e^{-\epsilon \cdot T + \mathcal{R}_i^{\mathrm{swap}}(T)}$$

Since there $n$ bidders and since there are at most $k^k$ different switching functions $\delta : \mathcal{S}_i \mapsto \mathcal{S}_i$ ($|\mathcal{S}_i| = k$), by union bound we get that ,

$$\Pr\left[\text{there exists agents } i \in [n] \text{ and } \delta : \mathcal{S}_i \mapsto \mathcal{S}_i \text{ with } \sum_{t=1}^{T} U_i(\delta(s_i^t), s_{-i}^t) - \sum_{t=1}^{T} U_i(s_i^t, s_{-i}^t) \geq \epsilon \cdot T\right] \leq \delta.$$

after at most $T = \mathcal{O}\left(\max_{i \in [n]} \mathcal{R}_i(T)/\epsilon + \log(n/\delta)/\epsilon + k \log k/\epsilon\right)$ rounds. Thus, with probability at least $1 - \delta$, the following holds for each bidder $i \in [n]$ and switching function $\delta : \mathcal{S}_i \mapsto \mathcal{S}_i$.

$$\frac{1}{T} \sum_{t=1}^{T} U_i(\delta(s_i^t), s_{-i}^t) - \frac{1}{T} \sum_{t=1}^{T} U_i(s_i^t, s_{-i}^t) \leq \epsilon. \tag{18}$$

By the definition of $\hat{\mu} \in \Delta(\mathcal{S})$ as $\hat{\mu}(s) = \frac{1}{T} \cdot \sum_{t=1}^{T} \mathrm{I}[s_t = s]$ we get that Equation (18) admits the following equivalent form,

$$\mathbb{E}_{s_i \sim \hat{\mu}} U_i(\delta(s_i), s_{-i}^t) - \mathbb{E}_{s_i \sim \hat{\mu}} U_i(s_i, s_{-i}^t) \leq \epsilon. \tag{19}$$

and thus that $\hat{\mu} \in \Delta(\mathcal{S})$ is an $\epsilon$-approximate correlated equilibrium. $\square$

## D. Omitted Proofs of Section 3.1

**Lemma D.1.** *Let $\mu \in \Delta(\mathcal{S})$ be an $\epsilon$-approximate correlated equilibrium. Then $\sum_{s \in \mathcal{S}} \mu(s) \cdot r(s) \geq Z_{LP}^\star$.*

*Proof.* Since $\mu \in \Delta(\mathcal{S})$ then $\sum_{s \in \mathcal{S}} \mu(s) = 1$ and $\mu(s) \geq 0$ for all $s \in \mathcal{S}$. Notice that by Definition 2.2 we know that for any bidder $i \in [n]$ and switching function $\delta : \mathcal{S}_i \mapsto \mathcal{S}_i$,

$$\sum_{s_i} \sum_{s_{-i}} \mu(s_i, s_{-i}) \cdot (U_i(s_i, s_{-i}) - U_i(\delta(s_i), s_{-i})) \geq -\epsilon$$

Let $s_i, s_i' \in \mathcal{S}_i$ and consider the switching function $\delta(s_i) = s_i'$ and $\delta(x) = x$ otherwise. Then we directly get that

$$\sum_{s_{-i}} \mu(s_i, s_{-i}) \cdot (U_i(s_i, s_{-i}) - U_i(s_i', s_{-i})) \geq -\epsilon \quad \text{for all } s \in \mathcal{S}.$$

$\square$

**Lemma D.2.** *The dual of the LP in Definition 3.1 is the following:*

$$\begin{aligned}
\text{max} \quad & \mu - \epsilon \cdot \sum_{i \in [n]} \sum_{s_i \in \mathcal{S}_i} \sum_{s'_i \in \mathcal{S}_i} \lambda^i_{s_i s'_i} \\
\text{s.t.} \quad & \mu + \sum_{i \in [n]} \sum_{s'_i \in \mathcal{S}_i} \lambda^i_{s_i s'_i} \left( U_i(s_i, s_{-i}) - U_i(s'_i, s_{-i}) \right) \le r(s) \\
& \text{for all } s \in \mathcal{S} \\
& \lambda^i_{s_i s_{-i}} \ge 0 \quad \forall i \in [n], s_i, s_{-i} \in \mathcal{S}_i
\end{aligned}$$

*Proof.* By taking the Lagragian

$$L := \sum_s \mu(s) \cdot r(s) - \sum_{i, s_i, s'_i} \lambda^i_{s_i, s'_i} \left( \epsilon + \sum_{s_{-i}} \mu(s_i, s_{-i}) \cdot (U_i(s_i, s_{-i}) - U_i(s'_i, s_{-i})) \right) + \mu \left( 1 - \sum_s \mu(s) \right) - \sum_s k_s \cdot \mu(s)$$

where $\lambda^i_{s_i, s_{-i}}, k_s \ge 0$. By rearranging the terms we get that

$$L := \sum_s \mu(s) \left( r(s) + \sum_{s'_i} \lambda^i_{s_i, s'_i} (U_i(s'_i, s_{-i}) - U_i(s_i, s_{-i})) - \mu - k_s \right) + \mu - \epsilon \sum_{i, s_i, s'_i} \lambda^i_{s_i, s'_i}$$

By setting $r(s) - \sum_{i \in [n]} \sum_{s'_i \in \mathcal{S}_i} \lambda^i_{s_i, s'_i} (U_i(s_i, s_{-i}) - U_i(s'_i, s_{-i})) - \mu - k_s = 0$ we get that

$$\mu + \sum_{i \in [n]} \sum_{s'_i \in \mathcal{S}_i} \lambda^i_{s_i, s'_i} (U_i(s_i, s_{-i}) - U_i(s'_i, s_{-i})) \le r(s)$$

since $k_s \ge 0$. $\square$

**Lemma D.3.** *Let a bidding profile $s \in \mathcal{S}$ such that $s_2 = \max(s_1, \ldots, s_n)$ and $s_1 < s_2$ then $b_s \ge v_2 - 2/k$.*

*Proof.* Since $s_2 = \max(s_1, \ldots, s_n)$ and $s_1 < s_2$, we know that bidder 1 does not win the item. Thus $U_1(s_1, s_{-1}) = 0$. Since $s_2 = \max(s_1, \ldots, s_n)$ we know that bidder 2 is among the winners meaning that $U_2(s_2, s_{-2}) \le v_2 - s_2$. The latter is due to the fact that in case of a tie, the item is given to one of the winner uniformly at random. Since $s_2 = \max(s_1, \ldots, s_n)$, we also know that if bidder 2 submits any bid $s'_2 \ge s_2 + \frac{1}{k}$, it will be the only winner and thus $U_2(s'_2, s_{-2}) = v_2 - s'_2$.

By Equation (2) in the proof of Lemma 3.8 we know that

$$\hat{b}_s = s_1 - \underbrace{\sum_{s'_1 = s_1 + 1/k}^{v_2 - 2/k} (U_1(s_1, s_{-1}) - U_1(s'_1, s_{-1}))}_{(A)} - \underbrace{\sum_{s'_2 = s_2 + 1/k}^{v_2 - 2/k} (U_2(s_2, s_{-2}) - U_2(s'_2, s_{-2}))}_{(B)}$$

Up next, we upper bound the terms (A) and (B) separately.

Starting with the term (B), we remind that $U_2(s'_2, s_{-2}) = v_2 - s'_2$ and $U_2(s_2, s_{-2}) \le v_2 - s_2$.

$$\begin{aligned}
(B) \quad := \quad & \sum_{s'_2 = s_2 + 1/k}^{v_2 - 2/k} (U_2(s_2, s_{-2}) - U_2(s'_2, s_{-2})) \le \sum_{s'_2 = s_2 + 1/k}^{v_2 - 2/k} ((v_2 - s_2) - (v_2 - s'_2)) \\
= \quad & \sum_{s'_2 = s_2 + 1/k}^{v_2 - 2/k} (s'_2 - s_2) = \frac{1}{k} \left( \frac{(kv_2 - 1)(kv_2 - 2)}{2} - \frac{ks_2(ks_2 + 1)}{2} \right) - s_2 \cdot (kv_2 - 2 - k \cdot s_2 - 1 + 1) \\
= \quad & \frac{1}{2k} \left( k^2 \cdot v_2^2 - 3k \cdot v_2 + 2 - k^2 \cdot s_2^2 - k \cdot s_2 \right) - \frac{1}{k} \left( k^2 \cdot s_2 v_2 - 2k \cdot s_2 - k^2 \cdot s_2^2 \right)
\end{aligned}$$

At the same time, since $s_1 < s_2 = \max(s_1, \ldots, s_n)$ we have that $U_1(s_1, s_{-1}) = 0$ (bidder 1 does not win the item). The same holds for all bids $s_1' \leq s_2 - 1$, meaning that $U_1(s_1', s_{-1}) = 0$. As a result, we get that

$$
\begin{aligned}
(\mathrm{A}) \quad := \quad & \sum_{s_1'=s_1+1/k}^{v_2-2/k} (U_1(s_1, s_{-1}) - U_1(s_1', s_{-1})) \\
= \quad & \sum_{s_1'=s_1+1/k}^{s_2-1} \left( \underbrace{U_1(s_1, s_{-1})}_{0} - \underbrace{U_1(s_1', s_{-1})}_{0} \right) + \sum_{s_1'=s_2}^{v_2-1/k} \left( \underbrace{U_1(s_1, s_{-1})}_{0} - U_1(s_1', s_{-1}) \right) \\
= \quad & -U_1(s_2, s_{-1}) - \sum_{s_1'=s_2+1/k}^{v_2-2/k} \underbrace{U_1(s_1', s_{-1})}_{\geq v_2-s_1'} \quad \text{since agent 1 wins the item} \\
\leq \quad & -\underbrace{U_1(s_2, s_{-1})}_{\leq 0} - \sum_{s_1'=s_2+1/k}^{v_2-2/k} (v_2 - s_1') \\
\leq \quad & -v_2(k \cdot v_2 - 2 - k \cdot s_2 - 1 + 1) + \frac{1}{k} \left( \frac{(kv_2 - 1)(kv_2 - 2)}{2} - \frac{ks_2(ks_2 + 1)}{2} \right) \\
\leq \quad & -\frac{1}{k}(k^2 \cdot v_2^2 - 2v_2 k - k^2 \cdot s_2 v_2) + \frac{1}{2k} \left( k^2 \cdot v_2^2 - 3k \cdot v_2 + 2 - k^2 \cdot s_2^2 - k \cdot s_2 \right)
\end{aligned}
$$

Putting everything together we get that

$$
\begin{aligned}
& \sum_{s_1'=s_1+1}^{v_2-2/k} (U_1(s_1, s_{-i}) - U_1(s_1', s_{-i})) + \sum_{s_2'=s_2+1}^{v_2-2/k} (U_2(s_1, s_{-i}) - U_2(s_2', s_{-i})) \\
\leq \quad & \frac{1}{k} \left( k^2 \cdot v_2^2 - 3k \cdot v_2 + 2 - k^2 \cdot s_1^2 - k \cdot s_1 \right) - \frac{1}{k} \left( k^2 \cdot s_1 v_2 - 2k \cdot s_1 - k^2 \cdot s_1^2 \right) \\
& - \frac{1}{k}(k^2 \cdot v_2^2 - 2k \cdot v_2 - k^2 \cdot s_1 v_2) \\
= \quad & -v_2 + \frac{2}{k} + s_1
\end{aligned}
$$

We thus finally get that

$$
\begin{aligned}
\hat{b}_s \quad = \quad & s_1 - \sum_{s_1'=s_1+1/k}^{v_2-2/k} (U_1(s_1, s_{-1}) - U_1(s_1', s_{-1})) - \sum_{s_2'=s_2+1/k}^{v_2-2/k} (U_2(s_2, s_{-2}) - U_2(s_2', s_{-2})) \\
\geq \quad & s_1 + v_2 - \frac{2}{k} - s_1 = v_2 - \frac{2}{k}
\end{aligned}
$$

$\square$

## D.1. Proof of Lemma 3.11

In this section we provide the proof of Lemma 3.11.

**Lemma D.4.** *Let a bidding profile* $s \in \mathcal{S}$ *such that* $s_2 = \max(s_1, \ldots, s_n)$ *and* $s_1 = s_2$ *then* $b_s \geq v_1 - 2/k$.

*Proof.* By Equation (2) in the proof of Lemma 3.8 we get

$$
\begin{aligned}
\hat{b}_s \quad = \quad & s_1 - \underbrace{\sum_{s_1'=s_1+1/k}^{v_2-2/k} \left( U^1(s_1, s_{-1}) - U^1(s_1', s_{-1}) \right)}_{(\mathrm{A})} \\
& - \underbrace{\sum_{s_2'=s_2+1/k}^{v_2-2/k} \left( U^2(s_2, s_{-2}) - U^2(s_2', s_{-2}) \right)}_{(\mathrm{B})}
\end{aligned}
$$

Since $s_2 = s_1$, by rearranging the terms we get $(A) + (B) =$

$$\sum_{s_1'=s_1+1/k}^{v_2-2/k} \left( \underbrace{(U^1(s_1, s_{-1}) + U^2(s_1, s_{-2}))}_{\leq v_1-s_1} - \underbrace{U^1(s_1', s_{-1})}_{=v_1-s_1'} \right) \quad (I)$$

$$- \sum_{s_2'=s_1+1/k}^{v_2-2/k} \underbrace{U^2(s_2', s_{-2})}_{=v_2-s_2'} \quad (II)$$

In Equation (I), $U^1(s_1, s_{-1}) + U^2(s_1, s_{-2}) \leq v_1 - s_1$ is due to the fact that since both bidder 1 and 2 submit the exact same bid $s_1 = \max(s_1, \ldots, s_n)$, then there are least 2 potential winners. As result, bidders 1 and 2 win with probability at most $1/2$. Thus,

$$U^1(s_1, s_{-1}) \leq (v_1 - s_1)/2 \text{ and } U^2(s_1, s_{-2}) \leq (v_2 - s_1)/2$$

which means that their sum is at most $v_1 - s_1$ since $v_2 \leq v_1$.

At the same time, $U_1(s_1', s_{-1}) = v_1 - s_1'$ comes from the fact that bidder 1 is the only winner if it submits any bid $s_1' \geq s_1 + 1/k$. For the exact same reason $U^2(s_2', s_{-2}) = v_2 - s_2'$ for $s_2' \geq s_1 + 1/k$ in Equation (5). As a result,

$$
\begin{aligned}
(A) + (B) &\leq \sum_{s_1'=s_1+1/k}^{v_2-2/k} (s_1' - s_1) - \sum_{s_1'=s_1+1/k}^{v_2-2/k} (v_2 - s_2') \\
&= 2 \cdot \sum_{s_1'=s_1+1/k}^{v_2-2/k} s_1' - \sum_{s_1'=s_1+1/k}^{v_2-2/k} (s_1 + v_2) \\
&= \frac{2}{k} \left( \frac{(kv_2-2)(kv_2-1)}{2} - \frac{(ks_1+1)(ks_1)}{2} \right) \\
&\quad - (s_1 + v_2)(kv_2 - 2 - ks_1) \\
&= kv_2^2 - 3v_2 + \frac{2}{k} - ks_1^2 - s_1 - ks_1 v_2 + 2s_1 \\
&\quad + ks_1^2 - kv_2^2 + 2v_2 + kv_2 s_1 \\
&= -v_2 + \frac{2}{k} + s_1
\end{aligned}
$$

Thus, $\hat{b}_s = s_1 - (A) - (B) \geq v_2 - \frac{2}{k}$. □

### D.2. Proof of Lemma 3.12

**Lemma D.5.** *Let a bidding profile $s \in S$ such that $s_1 \neq \max(s_1, \ldots, s_n)$ and $s_2 \neq \max(s_1, \ldots, s_n)$[7] then $b_s \geq v_2 - 3/k$.*

*Proof.* To simplify notation we denote with $s^\star := \max(s_1, \ldots, s_n)$ and $i^\star := \operatorname{argmax}(s_1, \ldots, s_n)$. Notice that $s^\star \leq v_{i^\star} \leq v_2$. For any bid $s \leq s^\star - 1/k$ we have that $U_1(s, s_{-1}) = U_2(s, s_{-2}) = 0$ since neither agent 1 nor agent 2 wins the item. At the same time $U_1(s_1', s_{-1}) = v_1 - s_1'$ for all $s_1' \geq s^\star + 1$. Similarly $U_2(s_2', s_{-2}) = v_2 - s_2'$ for all $s_2' \geq s^\star + 1$. As a result,

$$
\begin{aligned}
\hat{b}_s &= r_s - \sum_{i \in \{1,2\}} \sum_{s_i'=s_i+1/k}^{v_2-2/k} (U_i(s_i, s_{-i}) - U_i(s_i', s_{-i})) \\
&= s^\star - \sum_{i \in \{1,2\}} \left( \sum_{s_i'=s_i+1/k}^{s^\star-1} \left( \underbrace{U_i(s_i, s_{-i})}_{0} - \underbrace{U_i(s_i', s_{-i})}_{0} \right) + \sum_{s_i'=s^\star}^{v_2-2/k} \left( \underbrace{U_i(s_i, s_{-i})}_{0} - U_i(s_i', s_{-i}) \right) \right) \\
&= s^\star + \sum_{i \in \{1,2\}} \sum_{s_i'=s^\star}^{v_2-2/k} U_i(s_i', s_{-i}) \geq s^\star + \sum_{i \in \{1,2\}} \sum_{s_i'=s^\star+1}^{v_2-2/k} U_i(s_i', s_{-i}) \\
&= s^\star + \sum_{s_1'=s^\star+1}^{v_2-2/k} (v_1 - s_1') + \sum_{s_2'=s^\star+1}^{v_2-2/k} (v_2 - s_2') \geq s^\star + 2 \sum_{s_2'=s^\star+1}^{v_2-2/k} (v_2 - s_2')
\end{aligned}
$$

---

[7]Notice that the fact that $v_1 \geq v_2 \geq \ldots \geq v_n$ does imply that the bids $s_1, s_2, \ldots, s_n$ follow the same order.

As a result,

$$
\begin{aligned}
\hat{b}_s &\geq s^\star + 2 \sum_{s'_2 = s^\star + 1/k}^{v_2 - 2/k} (v_2 - s'_2) \\
&= s^\star + 2v_2(kv_2 - 2 - ks^\star + 1 - 1) - 2\frac{1}{k}\left(\frac{(kv_2 - 2)(kv_2 - 1)}{2} - \frac{ks^\star(ks^\star + 1)}{2}\right) \\
&= s^\star + 2kv_2^2 - 4v_2 - 2k \cdot s^\star v_2 - \frac{1}{k}(k^2 v_2^2 - 3kv_2 + 2 - k^2 s^{\star 2} - ks^\star) \\
&= s^\star + 2kv_2^2 - 4v_2 - 2k \cdot s^\star v_2 - kv_2^2 + 3v_2 - \frac{2}{k} + ks^{\star 2} + s^\star \\
&= 2s^\star + kv_2^2 - v_2 - 2k \cdot s^\star v_2 - \frac{2}{k} + ks^{\star 2} \\
&= 2s^\star + k(v_2 - s^\star)^2 - v_2 - \frac{2}{k}
\end{aligned}
$$

**Proposition D.6.** *If $v_2 \geq s^\star$ then $2s^\star + k(v_2 - s)^2 - v_2 \geq v_2 - \frac{1}{k}$.*

*Proof.* We will prove that $A := 2s^\star + k(v_2 - s^\star)^2 - 2v_2 \geq -\frac{1}{k}$. Notice that

$$
A = 2(s^\star - v_2) + k(s^\star - v_2)^2 = (s^\star - v_2)(2 + k(s^\star - v_2))
$$

In case $s^\star = v_2$ then $A = 0$. In case $s^\star = v_2 - \frac{1}{k}$ then $A = -\frac{1}{k}$. We conclude with the case where $s^\star \leq v_2 - \frac{2}{k}$. The latter implies that $2 + k(s^\star - v_2) \leq 0$ which in turn implies that $A \geq 0$. $\square$

Up next we conclude with the proof of Lemma 3.12.

$$
\hat{b}_s \geq 2s^\star + k(v_2 - s^\star)^2 - v_2 - \frac{2}{k} \geq v_2 - \frac{3}{k} \qquad \text{by Proposition D.6}
$$

$\square$

# E. Experimental Evaluations in the Bayesian Setting

In the section we experimentally evaluate the revenue produced by no-swap regret dynamics in the Bayesian setting where the valuation of each bidder $i \in [n]$ is sampled according to a probability distribution $\mathcal{P}_i \in \Delta(\mathcal{B})$. We consider the i.i.d. setting where each bidder $i \in [n]$ admits either $v_i = 0$ or $v_i = 1$ with probability $1/2$. In case of $n$ bidders then expected second highest valuation admits $\mathbb{E}[v_2] = 1 - (n+1)/2^{-n}$. We evaluate the algorithm of (Blum & Mansour, 2007) for $n = \{2, 3, 4, 5\}$ and $k = 10$. We remark that in the Bayesian setting each bidder runs on parallel two different no-swap regret algorithms - one responsible for the bids in case $v_i = 0$ and one responsible for the bids in case $v_i = 1$. In the following figure, we present our experimental findings indicating that even in the Bayesian setting the revenue produced converges to $\mathbb{E}[v_2]$.

We additionally examine the convergence properties of no-swap regret dynamics to Bayes Mixed Nash Equilibrium for $n = 2$ and i.i.d. valuations uniform in $\{0, 1\}$ (exactly as above). In case of continuous bidding space $\mathcal{B} = [0, 1]$, there exists a unique Bayes Mixed Nash Equilibrium $s^\star$ where $s^\star(0) = 0$ and $s^\star(1) \sim \mathcal{P}$ with CDF, $F_{\mathcal{P}}(x) = 1/(1-x) - 1$ for all $x \in [0, 1/2]$. In the following figure, we present the time-average bidding behavior of bidder 1 (the behavior of bidder 2 is almost identical) for $k = 50$ and $T = 10^4$ rounds. Our experimental findings reveal that the time-average bidding behavior of the bidders is very close to Bayes Mixed Nash Equilibrium. In particular for $v_i = 0$, bidder $i$ always bids 0 while for $v_i = 1$ bidder $i$ randomly selects its bid according to a CDF that closely approximates $F_{\mathcal{P}}(x) = \frac{1}{1-x} - 1$ for $x \in [0, 1/2]$.

We then repeat the same experiment for the more complex case where the valuation $v_i$ of each bidder $i \in \{1, 2\}$ is 0 with probability 0.25, 0.5 with probability 0.25, and 1 with probability 0.5. In this case $\mathcal{B} = [0, 1]$, then the Bayes Mixed Nash Equilibrium $s^\star(v_i)$ would be $s^\star(0) = 0$, $s^\star(0.5) \sim \mathcal{P}_{0.5}$, where $F_{\mathcal{P}_{0.5}}(x) = 1/(1-2x) - 1$ for $x \in [0, 0.25]$, and $s^\star(1) \sim \mathcal{P}_1$, where $F_{\mathcal{P}_1}(x) = 3/(4-x) - 1$ for $x \in [0.25, 5/8]$. In the following figure, we present our findings for $k = 30$ and $T = 50000$.

As we can see, not only the time-average revenue converges approximately to the expected second highest valuation ($\simeq 0.4026$), but also the respective distribution converges to the Bayes Mixed Nash Equilibrium.

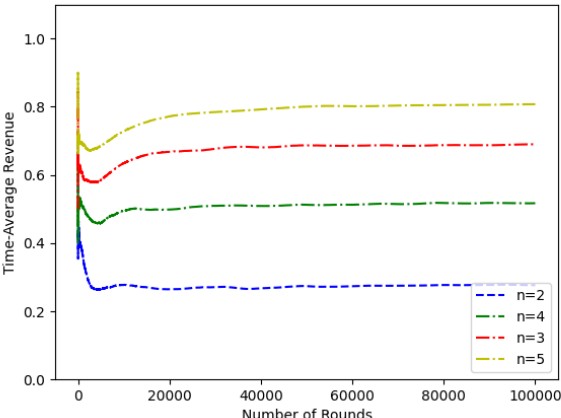

*Figure 4.* Revenue produced for different number of bidders.

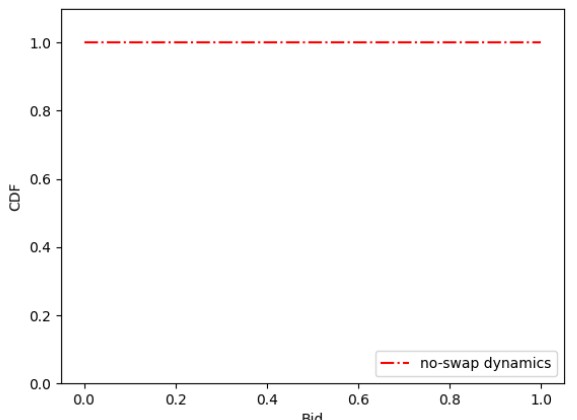

Time-averaged CDF played by bidder 1 once $v_1 = 0$.

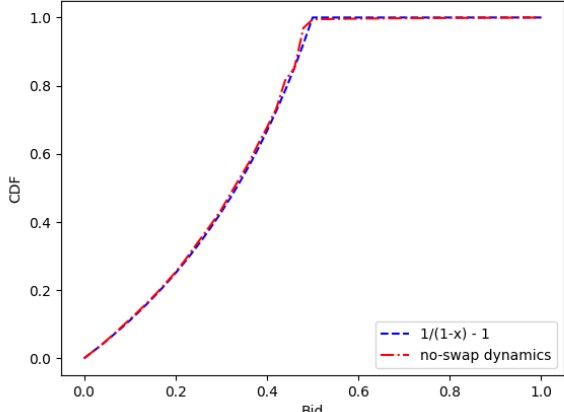

Time-averaged CDF played by bidder 1 once $v_1 = 1$.

*Remark* E.1. *In (Bichler et al., 2023), the authors provide interesting experimental evaluations showing that, in the case of a continuous bidding space ($\mathcal{B} = [0, 1]$), online learning dynamics converge to the unique Bayes Pure Nash Equilibrium. It is well known that when the bidding space is continuous and the distribution over bidders' valuations is also continuous, there exists a unique Bayes Pure Nash Equilibrium $s^\star : [0, 1] \to [0, 1]$. This implies that $s^\star(v)$ corresponds to a deterministic value in $[0, 1]$, which is why it is referred to as the Bayes Pure Nash Equilibrium. (Bichler et al., 2023) experimentally demonstrate that if bidders discretize their bidding space according to some discretization parameter $k \in \mathbb{N}$ and adopt online dual averaging algorithms, their strategies approximately converge to the Bayes Pure Nash Equilibrium $s^\star$.*

*However there are qualitative differences if the bidders retain continuous bidding space $\mathcal{B} = [0, 1]$ but the respective distribution has finite support $\mathcal{B}' \subseteq \mathcal{B}$. In this case, there exists a Bayes Mixed Nash Equilibrium in which each $s^\star(v)$ for all $v \in \mathcal{B}'$ is a continuous distribution over $[0, 1]$. As a result, the key difference between the experimental findings of (Bichler et al., 2023) and ours is that (Bichler et al., 2023) show convergence to a Bayes Pure Nash Equilibrium if valuations follow a continuous distribution over $[0, 1]$, whereas we demonstrate convergence to a Bayes Mixed Nash Equilibrium if valuations follow a probability distribution with discrete support in $[0, 1]$.*

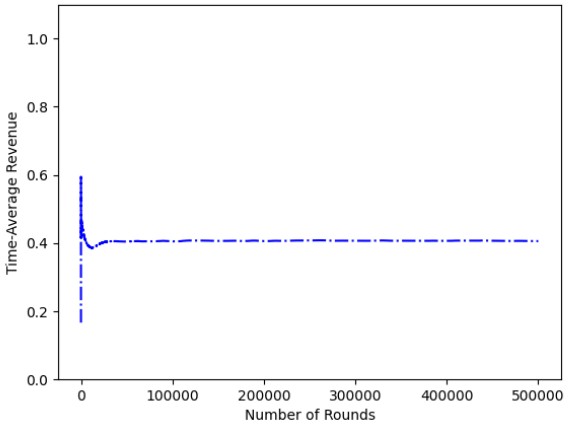

Time-average produced revenue.

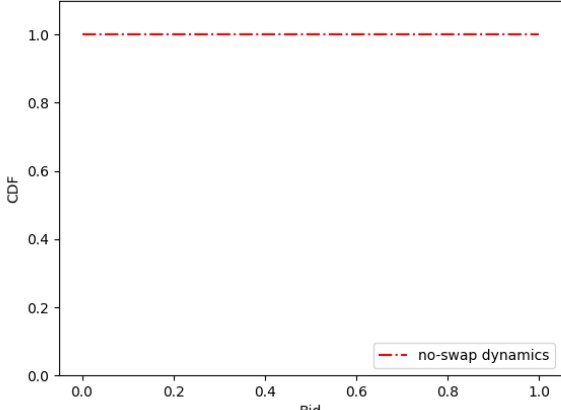

Time-averaged CDF played by bidder 1 once $v_1 = 0$.

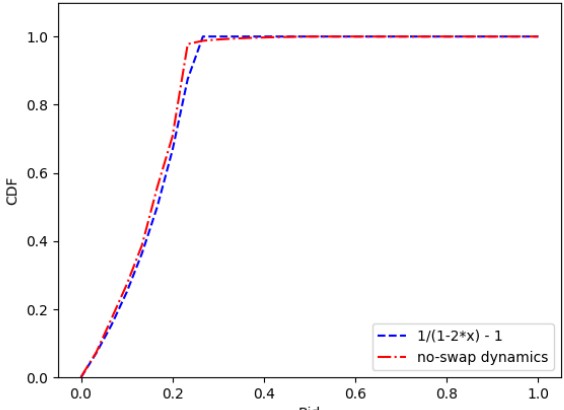

Time-averaged CDF played by bidder 1 once $v_1 = 0.5$.

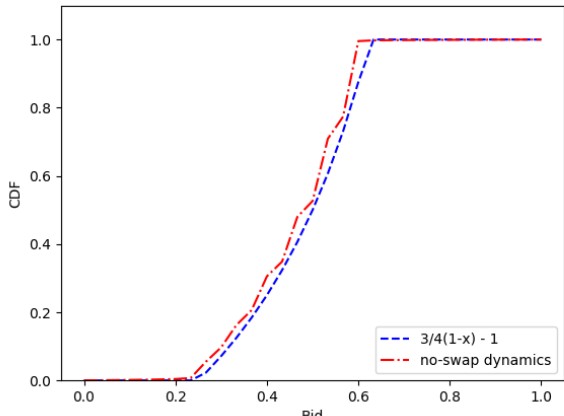

Time-averaged CDF played by bidder 1 once $v_1 = 1$.

