# OpenReview forum: "Revenue Guarantees of No-Swap-Regret Dynamics in First Price Auctions"
_ICML.cc/2026/Conference — ICML 2026 spotlight_

### Official Review · Reviewer_nHsF · 2026-03-05

**Soundness:** 3
**Presentation:** 3
**Significance:** 2
**Originality:** 3
**Overall Recommendation:** 4
**Confidence:** 3

**Summary:**

This paper looks at the revenue efficiency of correlated equilibria in first-price auctions. The setup here is that there are $n$ bidders and a set of bids that the bidders can replace (they assume that the bids are discretized). In contrast, prior work usually assumes that the bids are continuous whereas in practice, the granularity may be at 1 cent or something like that.

For the discrete case, one probably would expect that the optimal revenue is something like the second-highest bid minus the discretization error (they assume uniform discretization here so the error is $O(1/k)$). This turns out to be more or less their main result, if $\epsilon$-correlated equilibria is achieved then the error is $v_2 - \Theta(1/k) - \Theta(\epsilon k^2)$.

Since we know bounds on the regret, this also allows us to compute how many rounds it takes to get the $\epsilon$ term to be negligible thereby getting very close to the revenue of the correlated equilibria.

The authors also compare this with the arguments in prior papers which they argue would give only a quasi-polynomial error.

The authors also give some experimental evidence of their result.

**Compliance With Llm Reviewing Policy:**

Affirmed.

**Final Justification:**

I like the direction of the paper and that it makes progress in understanding correlated equilibria with discretization which is practical. However, a reservation I still have is that the paper makes no conclusion about whether there is a fundamental gap between discretization or whether it approaches the continuous case as the discretization gap decreases. So the message still feels incomplete.

Having said that, I still lean positive so will retain my score.

**Key Questions For Authors:**

1. Lots of mixing references. Please fix.
2. Are there provable gaps between the discrete case and the continuous case? As $k \to \infty$ with the bids constrained to $[0, 1]$, is the dependence on $k$ next to the $\epsilon$ necessary?

**Limitations:**

yes

**Strengths And Weaknesses:**

**Soundness.** The paper looks sound to me. The arguments seem quite interesting especially using the dual-fitting framework here, though I do not know much about that. The proofs appear to be non-trivial.

**Presentation.** The presentation is okay but there are a lot of undefined references. These should be fixed before the camera-ready or the next revision.

**Significance.** The results are interesting to me and I think will be of interest to others working in no-regret learning and mechanism design. For me, the main technical gap is that it is unclear if their results are tight. If the authors were able to prove that the dependence on $k$ next to the $\epsilon$ is necessary, i.e. one cannot do better than $v_2 - \Theta(1/k) - \Theta(\epsilon k^2)$ then I think that would make the result very interesting. Right now, we have a positive result for the discrete setting but the message of whether there is a real gap / issue between the discrete case and continuous case is unclear (for example, take $k \to \infty$ but just assume bids lie in $[0, 1]$). So the story feels a bit incomplete.

**Originality.** I have no issue with originality. This is an original problem and I think it is important to understand the impact of discretization.

---

> ### Author Rebuttal · Authors · 2026-03-26
>
> **Message to all reviewers.** We apologize for the broken references. The issue was caused by a synchronization conflict of the bibliography file that appeared in our very last submission, which we only noticed after the submission deadline. We apologize and have already taken care of it.
>
> ---
>
> Thank you for the positive review and insightful questions. We will address the missing references and other presentation issues in the camera-ready version.
>
> **Discrete case.** Our work focuses on the discrete-bid case, as in most practical settings there is some level of discretization in allowable bids. Our results leave an open gap between $\epsilon \cdot k^2$ and $\epsilon$, which we agree is a very interesting open question. We note, however, that our work already provides an exponential improvement over the previous $\epsilon \cdot k^{O(k)}$ bound, and contributes in opening the line of research to understanding the effect of discretization on the revenue guarantees.
>
> **Continuous case.** We completely agree that understanding whether an $\epsilon$-approximate CE in the continuous-bid case can guarantee revenue of at least $v_2 - O(\epsilon)$ is a very interesting question. We will highlight this as an interesting future direction in the conclusion section.
>
> **On gaps between discrete and continuous cases.** To the best of our knowledge, there are no provable gaps between the discrete and continuous cases. We conjecture that as $k \rightarrow \infty$, some dependence on $k$ in terms of $\epsilon$ is necessary; however, we believe this dependence should be linear rather than quadratic.
>
> We will incorporate the very interesting discussion above in the camera-ready version.

---

> > ### Author Rebuttal · Reviewer_nHsF · 2026-04-03
> >
> > Thanks for your response. I like the direction of the paper and that it makes progress in understanding correlated equilibria with discretization which is practical. However, a reservation I still have is that the paper makes no conclusion about whether there is a fundamental gap between discretization or whether it approaches the continuous case as the discretization gap decreases. So the message still feels incomplete.
> >
> > Having said that, I still lean positive so will retain my score.

---

### Official Review · Reviewer_2KDT · 2026-03-11

**Soundness:** 3
**Presentation:** 3
**Significance:** 3
**Originality:** 3
**Overall Recommendation:** 4
**Confidence:** 1

**Summary:**

The paper study the ϵ-approximate correlated equilibrium guarantees on the  revenue in first price auctions.
It shows that, if all bidders use an algorithm with no-swap regret  of at most regret $R(T)$, after $T$ rounds the revenue off the first price auction will be  $v_2−\Theta(1/k)−k^2\Theta(R(T)/T)$. Additionally, they show that to get a revenue of $v_2−\Theta(1/k)−\Theta(\epsilon)$, if the bidders use state-of-the-art swap-regret minimizing algorithm, only $T=O(k^5/\epsilon^2)$ rounds are necessary, characterizing the time needed to convergence to the $\epsilon$ optimal revenue up to a discretization error $\Theta(1/k)$.  The paper shows how the time-averaged revenue converges to the second-highest valuation in a number of rounds  polynomial in the number of discretized possible bids $(1/k)$. It does that by defining  an LP for which the optimal solution is less or equal than the value of the expected $\epsilon$-approximate correlated equilibrium, and study the LP through  its dual.

**Compliance With Llm Reviewing Policy:**

Affirmed.

**Key Questions For Authors:**

--

**Limitations:**

yes

**Strengths And Weaknesses:**

The research question seems quite interesting and the techniques and results seems well explained. Unfortunately I don't have enough familiarity with the literature to properly assess the true novelty of using a dual-fitting approach in this context,  which seems to be  a key factor to assess the actual contribution of the paper.  The idea seems however quite elegant and it seems to me well explained.

---

> ### Author Rebuttal · Authors · 2026-03-26
>
> **Message to all reviewers.** We apologize for the broken references. The issue was caused by a synchronization conflict of the bibliography file that appeared in our very last submission, which we only noticed after the submission deadline. We apologize and have already taken care of it.
>
> ---
>
> Thank you for the positive review and kind words. The dual-fitting approach is indeed a novel technique and is the key for obtaining an exponential improvement in the revenue bounds of approximate correlated equilibria.

---

> > ### Author Rebuttal · Reviewer_2KDT · 2026-04-04
> >
> > I would like to thank the authors for their answer, and I am inclined to maintain my original score.

---

### Official Review · Reviewer_qg32 · 2026-03-12

**Soundness:** 3
**Presentation:** 3
**Significance:** 3
**Originality:** 3
**Overall Recommendation:** 5
**Confidence:** 3

**Summary:**

This paper studies repeated first-price auctions with discrete price levels. Specifically, the paper attempts to improve lower bounds on the seller's revenue when the bidders play an approximate correlated equilibrium. The main result states that the revenue of any $\epsilon$-approximate correlated equilibrium is at least $v_2-\Theta(1/k)-\Theta(\epsilon k^2)$ where $v_2$ is the second-highest value of the bidders, which improves the prior bounds of $v_2-\Theta(1/k)-\epsilon\cdot k^{\mathtt{poly}(\log k)}$. The results further imply that the seller's revenue is at least $v_2-\Theta(1/k)-\epsilon \cdot \mathtt{poly}(k)$ when bidders use algorithms satisfying no-regret properties. The paper also conducts numerical experiments to validate its theoretical results.

**Compliance With Llm Reviewing Policy:**

Affirmed.

**Final Justification:**

The paper made a solid contribution to games with learning agents despite formating issue of the manuscript. I endorse the paper for acceptance. I had no concerns other than the formatting issues, and the rebuttal does not affect my assessment.

**Key Questions For Authors:**

No question needed.

**Limitations:**

Yes.

**Strengths And Weaknesses:**

## Strength
Although the dual-fitting techniques it uses are not entirely new, this paper qualitatively improves theoretical results from the previous literature on an interesting question in standard settings. The numerical experiment also supports the theoretical findings.

## Weakness
* Presentation
The draft is not very well-prepared, with a lot of minor issues, such as broken citations and section mismatches (Appendix 4 actually refers to Section 4)

---

> ### Author Rebuttal · Authors · 2026-03-26
>
> **Message to all reviewers.** We apologize for the broken references. The issue was caused by a synchronization conflict of the bibliography file that appeared in our very last submission, which we only noticed after the submission deadline. We apologize and have already taken care of it.
>
> ---
>
> Thank you for your positive review and interest in our techniques. We have already addressed the citation issues and section mismatches. In the camera-ready version, we will further correct typos and resolve any other minor issues.

---

> > ### Author Rebuttal · Reviewer_qg32 · 2026-03-31
> >
> > There was no significant concern that would affect the score. The formatting issue did not critically affect the paper's score.

---

### Official Review · Reviewer_mgBL · 2026-03-13

**Soundness:** 3
**Presentation:** 2
**Significance:** 3
**Originality:** 3
**Overall Recommendation:** 4
**Confidence:** 4

**Summary:**

This paper studies revenue guarantees in repeated discrete FPA when bidders use no-swap regret learning. They show that the time-average revenue converges to a level close to the second-highest value, and provide an improved polynomial bound on the number of rounds needed for this guarantee. Technically, the paper uses a primal-dual approach for analyzing approximate CE. They also provide numerical illustrations.

**Compliance With Llm Reviewing Policy:**

Affirmed.

**Final Justification:**

I am positive on this paper. I will keep my score as it is.

**Key Questions For Authors:**

Questions:
1. There are multiple missing-reference question marks throughout the paper, which leaves the reader with a poor impression of the paper's overall quality. In the informal theorem in Section 1.1, the concluding period is missing, and the placement of the footnote is potentially misleading.
2. Section 2.1 is overly verbose, as it reviews only the most basic material on first-price auctions and need not do so in such detail.
3. For readers who see only the title, a natural question is why the paper focuses on correlated equilibrium rather than Nash equilibrium. It would therefore be preferable for the title to make clear that the paper studies a dynamic first-price auction and investigates revenue efficiency under no-swap-regret algorithms.

**Strengths And Weaknesses:**

Strength:
The paper studies an important question in auction theory. The analysis is well organized, and the main argument is developed in a clear step-by-step manner. In addition, the paper's technical approach is fairly solid: the primal--dual framework employed in the analysis is well motivated and may prove useful in related settings.

Weakness:
There are simply far too many missing citations, which makes me question both the quality of the paper and the authors’ level of care.

---

> ### Author Rebuttal · Authors · 2026-03-26
>
> **Message to all reviewers.** We apologize for the broken references. The issue was caused by a synchronization conflict of the bibliography file that appeared in our very last submission, which we only noticed after the submission deadline. We apologize and have already taken care of it.
>
> ---
>
> Thank you for the positive feedback on our work.
>
> Addressing your specific questions and concerns
>
> 1. We have already fixed the citation issue and revised the informal theorem for the camera ready version.
> 2. We have taken this into account and will shorten section 2.1 in the camera ready version.
> 3. Good point, if ICML policy allows it we will update the title accordingly.

---

> > ### Author Rebuttal · Reviewer_mgBL · 2026-04-04
> >
> > Thank you for your detailed response. I appreciate the clarification and will keep my score as is.

---

### Decision · Program_Chairs · 2026-04-30

**Decision:**

Accept (spotlight)

**Comment:**

Overall the reviewers were positive about the paper, which makes several interesting contributions about revenue guarantees for approximate correlated equilibria in discrete first-price auctions. I agree with the reviewers’ assessment and recommend acceptance.

As a minor note, the reviewers highlighted some presentation issues that should be addressed in the final version of the paper.